A phylogenetic analysis of Bromus (Poaceae: Pooideae: Bromeae) based on nuclear ribosomal and plastid data, with a focus on Bromus sect. Bromus

http://orcid.org/0000-0001-7724-4837 Nasiri Akram 1 2 anasiri89@yahoo.com
Kazempour-Osaloo Shahrokh 1
Hamzehee Behnam 3
Bull Roger D. 2
http://orcid.org/0000-0003-1790-4332 Saarela Jeffery M. 2 jsaarela@nature.ca
1 Department of Plant Biology, Faculty of Biological Sciences, Tarbiat Modares University , Tehran , Iran
2 Beaty Centre for Species Discovery and Botany Section, Canadian Museum of Nature , Ottawa, Ontario , Canada
3 Botany Research Division, Research Institute of Forests and Rangelands, Agricultural Research, Education and Extension Organization (AREEO) , Tehran , Iran
Sosa Victoria
Electronic publication date: 2022 Sep 28
Publication date: 2022
Volume: 10
Electronic Location ID: e13884
Received 2022 Jan 7; Accepted 2022 Jul 21
Copyright: © 2022 Nasiri et al.
Copyright year: 2022
Copyright holder: Nasiri et al.
License: This is an open access article distributed under the terms of the Creative Commons Attribution License, which permits unrestricted use, distribution, reproduction and adaptation in any medium and for any purpose provided that it is properly attributed. For attribution, the original author(s), title, publication source (PeerJ) and either DOI or URL of the article must be cited.
License URL: https://creativecommons.org/licenses/by/4.0/

Keywords: Bromus sect. Genea, Bromus pectinatus complex, Grasses, Molecular systematics, Classification, Bromus sect. Bromopsis, Bromus sect. Ceratochloa, Bromus sect. Nevskiella, Bromus sect. Boissiera, Taxonomy

Funding: Tarbiat Modares University Canadian Museum of Nature Ministry of Science, Research, and Technology of Iran Financial support for the laboratory work was provided by the research council of the Tarbiat Modares University through a Ph.D. student fellowship to Akram Nasiri and by the Canadian Museum of Nature through research funding to Jeffery M. Saarela. The Ministry of Science, Research, and Technology of Iran provided support to Akram Nasiri for a research exchange at the Canadian Museum of Nature. The funders had no role in study design, data collection and analysis, decision to publish, or preparation of the manuscript.

==============================
To investigate phylogenetic relationships among and within major lineages of Bromus, with focus on Bromus sect. Bromus, we analyzed DNA sequences from two nuclear ribosomal (ITS, ETS) and two plastid (rpl32-trnLUAG, matK) regions. We sampled 103 ingroup accessions representing 26 taxa of B. section Bromus and 15 species of other Bromus sections. Our analyses confirm the monophyly of Bromus s.l. and identify incongruence between nuclear ribosomal and plastid data partitions for relationships within and among major Bromus lineages. Results support classification of B. pumilio and B. gracillimus within B. sect. Boissiera and B. sect. Nevskiella, respectively. These species are sister groups and are closely related to B. densus (B. sect. Mexibromus) in nrDNA trees and Bromus sect. Ceratochloa in plastid trees. Bromus sect. Bromopsis is paraphyletic. In nrDNA trees, species of Bromus sects. Bromopsis, Ceratochloa, Neobromus, and Genea plus B. rechingeri of B. sect. Bromus form a clade, in which B. tomentellus is sister to a B. sect. Genea–B. rechingeri clade. In the plastid trees, by contrast, B. sect. Bromopsis species except B. tomentosus form a clade, and B. tomentosus is sister to a clade comprising B. sect. Bromus and B. sect. Genea species. Affinities of B. gedrosianus, B. pulchellus, and B. rechingeri (members of the B. pectinatus complex), as well as B. oxyodon and B. sewerzowii, are discordant between nrDNA and plastid trees. We infer these species may have obtained their plastomes via chloroplast capture from species of B. sect. Bromus and B. sect. Genea. Within B. sect. Bromus, B. alopecuros subsp. caroli-henrici, a clade comprising B. hordeaceus and B. interruptus, and B. scoparius are successive sister groups to the rest of the section in the nrDNA phylogeny. Most relationships among the remaining species of B. sect. Bromus are unresolved in the nrDNA and plastid trees. Given these results, we infer that most B. sect. Bromus species likely diversified relatively recently. None of the subdivisional taxa proposed for Bromus sect. Bromus over the last century correspond to natural groups identified in our phylogenetic analyses except for a group including B. hordeaceus and B. interruptus.

Introduction

Bromus L. (Poaceae: Pooideae: Bromeae; conserved type B. secalinus L.) is a genus of annual, biennial, and perennial grasses, with 160 to 170 species widely distributed in temperate regions (Acedo & Llamas, 2001; Saarela et al., 2007; Saarela, Peterson & Valdés-Reyna, 2014). Several Bromus species are important forage grasses (Ferdinandez, Somers & Coulman, 2001; Puecher et al., 2001; Araghi et al., 2014) and many are invasive weeds (Ainouche et al., 1999; Otfinowski, Kenkel & Catling, 2007; Fink & Wilson, 2011; Huttanus, Mack & Novak, 2011). The following combination of characters distinguishes Bromus from other grass genera: sheaths connate for most of their length, awns subterminal when present, ovary apices with hairy bilabiate appendages, and simple starch grains (Smith, 1970). The genus has considerable variation in chromosome size, genome size, ploidy type, and ploidy level, which ranges from 2n = 14 to 2n = 112, with a base chromosome number of x = 7 (Stebbins, 1981; Armstrong, 1991; Tuna, Vogel & Arumuganathan, 2005, 2006; Klos et al., 2009; Sadeghian, Jafari & Hatami, 2010; Keshavarzi, Direkvandi & Khoshnood, 2016; Artico et al., 2017). Hybridization has played a role in the genus’s diversification and evolution.

Phylogenetic position of Bromus

Bromus is the sole genus in the tribe Bromeae (Soreng et al., 2017), and Bromeae and Triticeae, a tribe including important cereal species, are sister groups in most studies (e.g., Saarela et al., 2007, 2018; Fortune et al., 2008; Soreng et al., 2015; Hodkinson, 2018). In two studies based on whole plastomes, however, Bromeae is nested within Triticeae, rendering Triticeae paraphyletic (Bernhardt et al., 2017; Orton et al., 2021). Several authors have included Littledalea Hemsl., a genus with four species distributed in high altitudes on the Tibetan Plateau and adjacent Central Asian mountains (Soreng et al., 2015), in Bromeae because of its morphological similarities with Bromus (e.g., Stebbins, 1981; Clayton & Renvoize, 1986; Tzvelev, 1989; Kellogg, 2015). In phylogenetic analyses, however, Littledalea and Bromeae + Triticeae are sister groups (Soreng, Davis & Voionmaa, 2007; Schneider, Winterfeld & Röser, 2012; Soreng et al., 2015; Orton et al., 2021; Zhang et al., 2022). Some authors include Littledalea in tribe Littledaleeae Soreng & J.I.Davis (Soreng et al., 2015, 2017). Others include Bromus and Littledalea within a broadly circumscribed Triticeae and classify them in subtribes Brominae Dumort. and Littledaleinae Röser (Schneider et al., 2009; Hochbach, Schneider & Röser, 2015).

Subdivisional classification of Bromus

Subdivisional classifications of Bromus have varied. Most contemporary authors recognize generic subdivisions at the sectional rank, namely B. sect. Boissiera (Hochst. ex Steud.) P.M.Sm., B. sect. Bromus, B. sect. Bromopsis Dumort. (syn. B. sect. Pnigma Dumort.), B. sect. Ceratochloa (P.Beauv.) Griseb., B. sect. Genea Dumort., B. sect. Mexibromus Saarela, P.M.Peterson & Valdés-Reyna, B. sect. Neobromus (Shear) Hitchc., B. sect. Nevskiella (V.I.Krecz. & Vved.) Tournay, B. sect. Penicillus Llamas & Acedo, B. sect. Sinobromus Keng f., and B. sect. Triniusia (Steud.) Nevski (e.g., Smith, 1970; Pavlick et al., 2003; Saarela et al., 2007; Saarela, 2008; Saarela, Peterson & Valdés-Reyna, 2014; Naderi & Rahiminejad, 2015; Naderi et al., 2016; Llamas & Acedo, 2019). Some authors recognize generic subdivisions as subgenera, namely B. subg. Bromus, B. subg. Ceratochloa (P. Beauv.) Hack., B. subg. Festucaria Roth, B. subg. Neobromus Shear, B. subg. Nevskiella (V.I.Krecz. & Vved.) V.I.Krecz. & Vved., B. subg. Stenobromus (Griseb.) Hack, and B. subg. Triniusia (Steud.) Pènzes, nom. illeg. (e.g., Stebbins, 1981; Acedo & Llamas, 1999; Llamas & Acedo, 2019). Some authors recognize multiple genera, namely Anisantha K.Koch, Boissiera Hochst. & Steud., Bromopsis (Dumort.) Fourr., Bromus, Ceratochloa P.Beauv., Nevskiella V.I.Krecz. & Vved., Trisetobromus Nevski, and Triniusa Steud. (Tzvelev, 1976; Spalton, 2004; Valdés et al., 2009; Stace, 2010). Phylogenetic analyses of DNA sequence data support the monophyly of B. sects. Boissiera, Ceratochloa, Genea, Mexibromus, Neobromus, Nevskiella, and Penicillus, whereas analyses have resolved as B. sect. Bromopsis as polyphyletic (Ainouche & Bayer, 1997; Saarela et al., 2007; Pourmoshir, Amirahmadi & Naderi, 2019). Bromus sect. Bromus (including B. sect. Triniusia) and B. sect. Genea are monophyletic in nrDNA trees but not in plastid trees (Saarela et al., 2007; Fortune et al., 2008). Relationships among many of these groups differ between plastid and nrDNA trees, indicating a complex evolutionary history (Saarela et al., 2007; Fortune et al., 2008; Pourmoshir, Amirahmadi & Naderi, 2019).

Bromus sect. Bromus

Bromus sect. Bromus consists of about 40 annual or biennial species native to Eurasia. Most grow in disturbed habitats or are closely associated with cultivated grasses (Scholz, 1970, 2008b). Researchers described several taxa of B. sect. Bromus from Europe in the latter twentieth century, many of which they considered of recent origin (Scholz, 1972, 1989, 1995, 1997, 2008a, 2008b; Sales & Smith, 1990; Acedo & Llamas, 1997). Some species are rare or extinct (Smith, 1972; Smith & Sales, 1993; Rich & Lockton, 2002). Many species are weedy and have been widely introduced to other continents (Stebbins, 1981; Planchuelo & Peterson, 2000; Pavlick & Anderton, 2007; Scholz, 2008b; Saarela, Peterson & Valdés-Reyna, 2014). Bromus sect. Bromus species have an annual or biennial habit, ovate, oblong, elliptic, or lanceolate spikelets that taper distally, and lemmas either lacking awns or having one to five (rarely six) awns (Smith, 1972; Naderi & Rahiminejad, 2015; Pourmoshir, Amirahmadi & Naderi, 2017). Species of B. sect. Bromus are mostly self-fertilizing, with varying degrees of outcrossing potential (Ainouche, Misset & Huon, 1995; Ainouche & Bayer, 1997). Based on chromosome data, Stebbins (1981) hypothesized that species classified in B. sects. Bromus, Boissiera, and Genea are derived from B. sect. Bromopsis and evolved through adaptation to agricultural conditions.

Polyploidy occurs in more than half the species of B. sect. Bromus and is mostly restricted to the tetraploid level (2n = 4x = 28) (Ainouche & Bayer, 1997; Scholz, 2008b), although triploids (2n = 3x = 21) and octoploids (2n = 8x = 56) have been recorded (Scholz, 2008b; Keshavarzi, Direkvandi & Khoshnood, 2016). Data from chromosomes (Stebbins, 1981; Armstrong, 1991; Ainouche, Misset & Huon, 1995), serology (Smith, 1972), isozymes (Ainouche, Misset & Huon, 1995; Oja, 1998), and DNA sequence data (Ainouche & Bayer, 1997; Ainouche et al., 1999) indicate that most tetraploids in the section are allopolyploids that arose via intrasectional hybridization. By contrast, researchers have hypothesized that B. arenarius Labill. and the B. pectinatus Thunb. complex arose via intersectional hybridization between species of B. sects. Bromus and Genea (Stebbins, 1956, 1981; Smith, 1972; Scholz, 1981).

The Bromus pectinatus complex

The Bromus pectinatus complex comprises annual tetraploid species ranging from southern Africa to Tibet. Scholz (1981) recognized six species in the complex: B. pectinatus, B. pulchellus Fig. & De Not. (syns. B. tytthanthus Nevski, B. sinaicus (Hack.) Täckh.), B. rechingeri Melderis, B. gedrosianus Pénzes, B. tibetanus H.Scholz, and B. pseudojaponicus H.Scholz. More recently, Naderi & Rahiminejad (2015) recognized four species in the complex: B. gedrosianus, B. pectinatus, B. pulchellus, and B. tibetanus. They treated the names B. rechingeri and B. pseudojaponicus as synonyms of B. pulchellus. Researchers classify species of the B. pectinatus complex in B. sect. Bromus, although they have morphological similarities with B. sect. Genea, such as cuneiform spikelets (Scholz, 1981; Sales, 1993). Scholz (1981) and Sales (1993) considered the B. pectinatus complex to blur the boundaries of B. sects. Bromus and Genea. Morphologically, B. pectinatus is intermediate between B. japonicus Thunb. (2n = 14; B. sect. Bromus) and B. tectorum L. (2n = 14; B. sect. Genea). Isozyme variation patterns in B. pectinatus, B. japonicus, and B. tectorum support the idea that B. pectinatus may have arisen from hybridization between B. japonicus and B. tectorum (Oja, 2007). Genetic data also support a putative hybrid origin of B. pectinatus involving species of B. sections Bromus and Genea (Saarela et al., 2007). Of the B. pectinatus complex species, researchers have included only B. pectinatus in DNA-based studies.

Phylogenetic relationships among species of Bromus section Bromus

Few studies have investigated phylogenetic relationships among B. sect. Bromus species with DNA sequence data. Ainouche & Bayer (1997) investigated the origins of tetraploid species using ITS sequences for 10 diploids and 12 tetraploid species. They recovered B. sect. Bromus as monophyletic, but with only three species representing other Bromus sections. In their tree including diploids and tetraploids, B. alopecuros subsp. caroli-henrici (Greuter) P.M.Sm., B. interruptus (Hack.) Druce–B. hordeaceus L., and B. scoparius L.–B. alopecuros Poir. subsp. alopecuros were successive sister groups to the rest of the section, and relationships among the remaining taxa were mostly poorly supported. Saarela et al. (2007) investigated the molecular phylogeny of Bromus using data from ITS and two plastid regions. Although they included few B. sect. Bromus species, they found B. danthoniae Trin. ex C.A.Mey., which many authors have classified in B. sect. Triniusia, to be nested within B. sect. Bromus. Additionally, they found that affinities of B. sects. Bromus and Genea differed between plastid and nrDNA trees. In their nrDNA trees, B. sect. Bromus and B. sect. Genea were monophyletic and not closely related, whereas in their plastid trees, species of these sections formed a clade and neither section was monophyletic. Fortune et al. (2008) reconstructed the origins of B. sect. Genea polyploids and included multiple B. sect. Bromus species in their analyses. Based on plastid data, they found B. sect. Bromus to be monophyletic in parsimony and maximum likelihood analyses and B. sect. Genea to be monophyletic in parsimony analyses. Based on ITS data, they found B. sect. Bromus and B. sect. Genea to be monophyletic. In their Waxy trees, neither B. sect. Genea nor B. sect. Bromus was monophyletic, as species of both sections were mixed in a clade. These previous studies represented B. sect. Bromus by few species and few molecular markers. Researchers have not studied the phylogeny of B. sect. Bromus with extensive taxon sampling, multiple individuals per taxon, and more than three DNA regions. Furthermore, researchers have not assessed subdivisional classifications of B. sect. Bromus (Table 1) in a phylogenetic context.

Table 1 Summary of subdivisional classifications of Bromus subg. Bromus/B. sect. Bromus proposed by different authors, including nomenclature, diagnostic characters as identified by the author, and species included in the taxon by the author.

Author	Classification	Diagnostic characters	Species included	
Holmberg (1924)	Bromus subg. Zeobromus (Griseb.) Hack. [=Bromus subg. Bromus]	B. sect. Macrantheri Holmb.	anthers ½ or more the length of the palea	B. arvensis L. (type), B. brachystachys Hornung	
		B. sect. Brachyantheri Holmb.	anthers 1/20–1/5(–1/3) the length of the palea		
		B. subsect. Coriacei Holmb., nom. superfl.	lemmas thick, ± leathery, and with nerves barely raised (palea inferiore crassior, ± coriacea, nervis vix elevates)	B. commutatus Schrad., B. japonicus, B. racemosus L., B. secalinus, B. squarrosus L.	
		B. subsect. Molles Holmb.	lemmas thin, membranous, often longitudinally pleated, and with nerves ± prominent (palea inferior tenuis, scariosa, saepe longitudinaliter plicata, nervis ± prominentibus)	B. hordeaceus (type, designated by Tournay, 1961), B. intermedius Guss., B. lanceolatus Roth, B. lepidus Holmb., B. mollis L. [=B. hordeaceus], B. scoparius	
Nevski (1934)	Bromus s.str.	B. sect. Aphaneuroneuron Nevski, nom. superfl.	lemma commonly with a short bifid awn below the apex (glumella infra apicem vulgo breviter bifidum arista unica plus minusve longararius fere obsolete donate)	B. arvensis, B. brachystachys, B. commutatus, B. japonicus, B. secalinus, B. squarrosus, B. racemosus	
		B. sect. Sapheneuron Nevski	lemma thin, scarious, acutely and widely forked at the apex, with more or less prominent nerves (glumella tenuis, scariosa, apice vulgo acute longeque bifida, nervis plus minusve prominentibus)	B. mollis [=B. hordeaceus], B. hordeaceus, B. intermedius, B. lanceolatus, B. lepidus, B. macrostachys Desf. [=B. lanceolatus] (type, designated by Scholz, 2008a), B. oxyodon Schrenk, B. scoparius, B. sewerzowii, B. tytthanthus	
		B. sect. Triniusia (Steud.) Nevski	lemma with tripartite awn (glumella infra apicem acute bipartitum triaristata)	B. danthoniae	
Kreczetovich & Vvedensky (1934)	B. subg. Zeobromus [=B. subg. Bromus]	B. series Secalini V.I.Krecz. & Vved., nom. superfl.	panicles erect, with long branches; spikelets oblong-conical, their lemmas not overlapping those of the adjacent row (the pedicels visible); lemmas subobtuse, awnless, or short-awned	B. secalinus	
		B. series Macrantherae V.I.Krecz. & Vved.	panicles spreading, with long branches; spikelets narrow, lanceolate; glumes, lemmas, and paleas lanceolate or oblong-lanceolate; and anthers 4 mm, linear	B. arvensis	
		B. series Squarrosae V.I.Krecz. & Vved.	inflorescences rather open; awns of the lower florets in a spikelet shorter than those of the upper florets, or spikelets awnless, anthers ca. 1 mm	B. anatolicus Boiss. & Heldr. [=B. japonicus subsp. anatolicus (Boiss. & Heldr.) Pénzes], B. briziformis Fisch. & C.A.Mey., B. japonicus, B. squarrosus	
		B. series Commutatae V.I.Krecz. & Vved.	rather contracted inflorescences, spikelets on short, mostly erect branches, awns of all florets in the spikelet equal, anthers ca. 2 mm	B. commutatus, B. mollis [=B. hordeaceus], B. popovii Drobow [=B. racemosus], B. racemosus	
		B. series Dolicholepides V.I.Krecz. & Vved.	inflorescences crowded, branches shorter than the spikelets; lemmas 7–10 mm, linear-lanceolate; awns straight or recurved, about as long as the lemmas; anthers 0.5–l mm	B. gracilis Popov, nom. nud. [=B. pulchellus], B. scoparius, B. sewerzowi [i.e., sewerzowii]	
		B. series Ambiguae V.I.Krecz. & Vved.	inflorescences spreading, with long, more or less divided branches; lemmas oblong-elliptical, 12–15 mm, with a slightly divergent awn, anthers 1 mm	B. oxyodon	
		B. series Macrostachyae V.I.Krecz. & Vved.	inflorescence contracted, branches straight, shorter than the spikelets; lemmas broadly elliptical, prominently nerved, awns ascending, anthers 1–2 mm	B. danthoniae, B. macrostachys [=B. lanceolatus]	
Pénzes (1936)	B. subg. Serrafalcus (Parl.) Pénzes [=B. subg. Bromus]	“B. sect. Arvenses”	–	B. arvensis, B. brachystachys, B. intermedius	
		“B. sect. Racemosi”	–	B. aegyptiacus Tausch, B. racemosus, B. tuzsonii Pénzes [=B. racemosus]	
		“B. sect. Commutati”	–	B. abolinii Drobow [=B. japonicus subsp. japonicus], B. briziformis, B. commutatus, B. japonicus, B. javorkae Pénzes [=B. hordeaceus subsp. hordeaceus], B. lepidus, B. macrostachys [=B. lanceolatus], B. mollis [=B. hordeaceus], B. oxyodon, B. secalinus, B. squarrosus	
		“B. sect Pectinati”	–	B. alopecuroides Poir., nom. illeg. superfl. [=B. alopecuros], B. degenii Pénzes [=B. scoparius], B. gedrosianus, B. pectinatus, B. scoparius, B. sewerzowii, B. szaboi Pénzes [=B. chrysopogon Viv.]	
Tournay (1961)	B. sect. Bromus	B. subsect. Coriacei, nom. superfl.	–	–	
		B. subsect. Triniusia (Steud.) Tournay	–	–	
		B. subsect. Molles	–	–	
Scholz (1970)	B. subg. Bromus	B. sect. Triniusia	–	–	
		B. sect. Bromus			
		B. series Intermediae H.Scholz	lemma lanceolate, awns spreading (lemma lanceolatum arista divaricato-patente)	B. intermedius (type)	
		B. series Macrantherae	anthers equal or exceeding half the length of lemma, lower sheaths covered with short appressed hairs (antherae longitudine paleam dimidiam aequantes vel superantes, vaginae inferiores pilis brevibus adpressis vestitae)	B. arvensis (type), B. brachystachys	
		B. series Squarrosae	panicles nodding, branches with 1–4 spikelets (panicula subnutans ramis 1–4 spiculatis)	B. japonicus [+ B. anatolicus (=B. japonicus subsp. anatolicus), B. briziformis, B. oostachys Bornm. (=B. lepidus)], B. squarrosus (type)	
		B. series Molles H.Scholz	panicle more or less contracted, lower sheath softly hairy (panicula plus minusve contracta, vaginae inferiores mollissime hirsutae)	B. hordeaceus (type), B. lepidus, B. molliformis Lloyd [=B. hordeaceus subsp. divaricatus (Bonnier & Layens) Kerguélen]	
		B. series Interruptae H.Scholz	palea bifid to the base (palea ad basim bifida)	B. interruptus (type)	
		B. series Racemosae H.Scholz	lateral parts of the palea smooth on the outside, lower sheath covered with rigid hairs (partes laterales paleae extus leaves, vaginae inferiores pilis patentibus rigidioribus obsitae)	B. pseudosecalinus P.M.Sm., B. racemosus (type)	
		B. series Secalinae, nom. superfl. [=B. series Bromus]	lateral parts of the palea prickly or rough; lemma leathery, nerves barely elevated (partes laterales paleae extus hirsutae vel scabrosae, lemma coriaceum nervis vix elevates)	B. commutatus, B. grossus Desf. ex DC. [+ B. neglectus (Parl.) Nyman (=B. commutatus subsp. neglectus (Parl.) P.M.Sm.), B. aegyptiacus, B. palaestinus (Melderis) Mouterde (=B. brachystachys), B. popovii (=B. racemosus)], B. secalinus (type)	
		B. series Michelariae H.Scholz	lemma leathery, dentate, nerves prominent (lemma coriaceum utrinque infra medium dento acutiusculo acutum, nervis prominentibus)	B. bromoideus (Lej.) Crép. (type)	
Smith (1972) 1	Bromus sect. Bromus	Group 1, A	lemmas papery in texture, broad, margins angled; panicle erect, dense; grains sometimes equalling or exceeding the paleas, obovate-oblanceolate; awns straight, erect, arising about 1/20–1/8 down the length of the lemma	B. hordeaceus, B. interruptus, B. lepidus, B. ×pseudothominei P.M.Sm.	
		Group 2, B	lemmas horny in texture, short, broad, margins angled; panicle lax, branches erect or spreading; grain shorter than palea, narrowly obovate-oblanceolate; awns straight, arising 1/20–1/8 down the length of the lemma	B. bromoideus, B. commutatus, B. grossus, B. pseudosecalinus (morphological classification), B. racemosus, B. secalinus	
		Group 3, C	lemmas horny in texture at least in fruit, short, broad, margins angled; panicle lax, branches spreading or drooping; grains sometimes equalling or exceeding the palea, elliptical to oblanceolate; awns usually straight, arising 1/20–1/8 down the length of the lemma	B. aegyptiacus (morphological classification), B. arvensis, B. brachystachys, B. palaestinus [=B. brachystachys] (morphological classification)	
		Group 4, E	lemmas horny in texture, often very broad, margins markedly angled; panicle lax, branches spreading or drooping; grain shorter than palea, obovate-oblanceolate; awns, when present, divaricate, arising about 1/4 down the length of the lemma	B. briziformis, B. japonicus, B. squarrosus	
		Group 5, H	lemmas papery in texture, short, fairly narrow, margins bluntly angled; panicle lax, branches erect or spreading; grains about equalling the paleas, narrowly elliptical; awns divaricate, arising about 1/4–1/3 down the length of the lemma	B. intermedius	
		Group 6, D	lemmas papery in texture, short, narrow, margins scarcely angled or rounded; panicles extremely dense or verticillate; grain shorter than palea, narrowly elliptical; awns divaricate, arising about 1/4 down the length of the lemma	B. scoparius	
		Group 7, F	lemmas papery in texture, long, usually narrow, margins angled; panicles erect, with stiff branches, open in flower but often densely contracted in fruit; grains shorter than palea, narrowly elliptical to linear; awns single or multiple, divaricate, sometimes twisted at the base, arising 1/4–1/3 down the length of the lemma	B. alopecuroides [=B. alopecuros], B. danthoniae, B. lanceolatus, B. pumilio (Trin.) P.M.Sm.	
		Group 8, G	lemmas papery in texture, long, very bluntly angled; panicles lax, branches spreading; grain shorter than palea, narrowly elliptical; awns divaricate, arising about 1/3 down the length of the lemma	B. oxyodon	
		Group 9, J	lemmas papery in texture, long, narrow, margins rounded or very bluntly angled; panicle lax, often drooping; grain shorter than palea, narrowly oblanceolate; awns straight, arising 1/5–1/4 down the length of the lemma	B. adoensis Hochst. ex Steud. [=B. pectinatus], B. arenarius, B. pectinatus	
		Group –, I	–	B. palaestinus [=B. brachystachys], B. pseudosecalinus (serology classification, both species)	
Tzvelev (1976)	Bromus s.str	B. sect. Aphaneuroneuron, nom. superfl. [=B. sect. Bromus]	lemmas oblongate-ovate or obovate, usually coriaceous, apex obtuse or with two distinct teeth, always with one awn, and spikelets slightly compressed laterally	B. arvensis, B. briziformis, B. commutatus, B. hordeaceus, B. japonicus, B. mollis [=B. hordeaceus], B. racemosus, B. secalinus, B. squarrosus	
		B. sect. Triniusia	lemmas oblongate-ovate or ovate, coriaceous, with two acute apical teeth and usually three awns, and spikelets highly compressed laterally	B. danthoniae	
		B. sect. Sapheneuron	lemmas oblongate-lanceolate, usually thin-coriaceous, with a broad membranous border and acute apical teeth, almost always single-awned, and spikelets slightly compressed laterally	B. gedrosianus, B. oxyodon, B. pseudodanthoniae Drobow [=B. danthoniae var. danthoniae], B. scoparius, B. sewerzowii, B. tytthanthus	
Tzvelev (1999)	Bromus s.str.	B. sect. Bromus	lemmas oblongate-ovate or obovate, usually coriaceous, apex obtuse or with two distinct teeth, always with one awn, and spikelets slightly compressed laterally	B. arvensis, B. briziformis, B. commutatus, B. hordeaceus, B. japonicus, B. mollis [=B. hordeaceus], B. racemosus, B. secalinus, B. squarrosus	
		B. sect. Triniusia	lemmas oblongate-ovate or ovate, coriaceous, with two acute apical teeth and usually three awns, and spikelets highly compressed laterally	B. danthoniae	
		B. sect. Sapheneuron	lemmas oblong-lanceolate, usually thin-coriaceous, with two acute teeth at apex, sometimes with an awn; spikelets somewhat flattened on sides	B. scoparius	
Acedo & Llamas (2005)		B. sect. Bromus	inflorescences panicles with erect or erect-patent branches; lemmas rhombic, with an awn inserted less than 2 mm from the apex and divaricate at maturity	B. arvensis, B. brachystachys, B. commutatus B. elidis H.Scholz, B. japonicus, B. lusitanicus Sales & P.M.Sm. [=B. racemosus fide], B. pseudobrachystachys H.Scholz, B. pseudosecalinus, B. racemosus, B. secalinus	
		B. sect. Squarrosi Acedo & Llamas	inflorescences racemose, secund, and pendent at maturity or at least lax and with few spikelets; lemmas broadly rhombic with apices bidentate even in the juvenile stages, muticous or with an awn inserted at least two mm from the apex and strongly divaricate at maturity	B. briziformis, B. psammophilus P.M.Sm., B. pseudobrachystachys, B. regnii H.Scholz [=B. japonicus subsp. japonicus fide Cabi & Doğan (2012)], B. squarrosus (type), B. tigridis Boiss. & Nöe [=B. brachystachys]	
Scholz (2008a)	Bromus s.str.	B. sect. Bromus	glumes and lemmas ± leathery, with no or hardly protruding nerves at maturity	B. brachystachys, B. commutatus, B. grossus, B. japonicus, B. oostachys [=B. lepidus], B. pseudosecalinus, B. racemosus, B. squarrosus	
		B. sect. Sapheneuron	glumes and lemmas with protruding nerves at maturity	B. hordeaceus, B. incisus R.Otto & H.Scholz, B. lepidus, B. macrostachys [=B. lanceolatus] (type, designated by Scholz, 2008a)	
		B. sect. Triniusia	–	B. danthoniae	
Notes:

En-dashes in the diagnostic characters and species included columns indicate the authors did not provide information. For diagnoses translated from Latin, the Latin text is included in parentheses.

1 Smith’s (1972) groups distinguished by numbers are based on morphological characteristics and groups distinguished by letters are based on protein serology data.

Objectives

We aimed to reconstruct phylogenetic relationships among and within major lineages of Bromus, with focus on Bromus sect. Bromus. We sequenced two nrDNA and two plastid regions from a broad species sampling with two or more individuals from most taxa to achieve the following objectives: (1) reconstruct phylogenetic relationships among major Bromus lineages; (2) characterize the phylogenetic affinities of the B. pectinatus complex; (3) reconstruct phylogenetic relationships among species within B. sect. Bromus; and (4) assess existing subdivisional classifications of B. sect. Bromus in a phylogenetic context.

Material and Methods

Taxon sampling

We sampled 103 in group specimens, including 87 (26 taxa) from B. section Bromus and 16 (15 species) from other Bromus sections, in the phylogenetic analyses. We obtained tissue samples from morphological vouchers deposited in herbaria at the Research Institute of Forests and Rangelands, Tehran, Iran (TARI), the University of Tabriz (HCAT), Tabriz, Iran, the University of Isfahan, Isfahan, Iran (HUI), the Canadian Museum of Nature, Ottawa, Canada (CAN), Naturhistorisches Museum Wien, Austria, Vienna (W), the University of Helsinki, Helsinki, Finland (H), and Tarbiat Modares University Herbarium, Tehran, Iran (TMUH), or from fresh material collected in 2016 in Iran by A. Nasiri and colleagues. AN and JMS determined or confirmed the identities of most specimens sampled, and Bromus expert R. Naderi had determined a subset of the specimens from Iran we sampled (e.g., Naderi et al., 2012; Naderi & Rahiminejad, 2015; Naderi et al., 2016). We also included sequence data for 23 specimens (22 species) retrieved from GenBank. We used six taxa from the tribes Triticeae and Littledaleeae as outgroups. Voucher information and GenBank accession numbers for new and previously published data are listed in Table 2.

Table 2 Voucher information for the samples used in the phylogenetic analyses of this study, including species name, geographical origin, collector(s), voucher (herbarium), and GenBank accession numbers.

Species		Collection information	GenBank accession No.	
ITS	ETS	matK	rpl32-trnLUAG	
Bromus alopecuros Poir.		Morocco: Tiznt, Jbel Imzi, Llamas & Gomiz 11 (LEB)	KM077300 1	KJ632450 1	×	×	
Bromus alopecuros subsp. caroli-henrici (Greuter) P.M.Sm.	(1)	Jordan: From seed, cultivated by R. Keane in Berkshire, UK, 28 August 2001, Keane RK007 (CAN 589846)	OM141017	OM033041	OM048521	OM056558	
	(2)	Israel: From seed, cultivated by R. Keane in Berkshire, UK, 28 August 2001, Keane RK003 (CAN 589842)	OM141018	OM033042	OM048522	OM056559	
Bromus arenarius Labill.		USA: Ferris et al. 13906	KP987314 2	KP996927 2	×	×	
Bromus arvensis L.	(1)	Iran: East Azerbaijan, 35 km to Maku from Marand, 28 June 1978, Assadi & Mozaffarian s.n. (TARI 30080)	OM141019	OM033043	OM048523	OM056560	
	(2)	Canada: Ontario, Ottawa, Experimental Farm, 9 July 1923, Malte s.n. (CAN 231110)	OM141020	OM033044	OM048524	OM056561	
	(3)	Canada: British Columbia, Invermere, 24 July 1915, Malte s.n. (CAN 124519)	OM141021	OM033045	OM048525	OM056562	
	(4)	Germany: accession 06-116-74-74 Botanischer Garten Oldenburg, 2011, Alonso 344 (LEB)	KM077301 1	KJ632451 1	×	×	
Germany, Hessen: C 618 (C)	×	×	HM770811 3	×	
Bromus berteroanus Colla		Peru: Bolognesi, Raquia- Cajacay, 2005, Peterson 17689 (MO) [This is the voucher information cited in Alonso et al. (2014). However, Peterson 17689 is Geranium sp., from Mexico (see http://n2t.net/ark:/65665/3b32d614c-0ad9-4332-bff1-f3d844b798db). The correct voucher information is likely Peru: Ancash, Bolognesi, 8 km E of Raquia & 2 km W of Cajacay on Ruta 02-014, 21 Mar 2004, Peterson et al. 17879 (see http://n2t.net/ark:/65665/372497496-be8f-4452-a7f8-abaa621b494d)]	KM077293 1	KJ632443 1	×	×	
Bromus brachystachys Hornung	(1)	Iran: Gilan, Talesh, Asalem, 16 June 1965, Esfandiari 10015E (TARI 23848)	OM141022	OM033046	×	OM056563	
	(2)	Germany: Sachsen-Anhalt, s.d., Hornung s.n. (W-Rchb. 1889-0230465)	×	×	OM048526	OM056564	
Bromus briziformis Fisch. & C.A.Mey.	(1)	Iran: Mazandaran, 27 km to Haraz road from Kandovan, 23 June 1979, Assadi & Mozaffarian s.n. (TARI 33093)	OM141023	OM033047	OM048527	OM056565	
	(2)	Iran: East Azerbaijan, Arasbaran protected area, Kaleybar toward Hejrandost, 7 June 1976, Assadi & Mozaffarian s.n. (TARI 20071)	×	×	×	OM056566	
	(3)	Iran: Mazandaran, Kelardasht, Rodbarak, 10 June 1973, Fotovat s.n. (TARI 10180)	OM141024	OM033048	OM048528	OM056567	
	(4)	Iran: Mazandaran, Sari, Vavsar village, S slope of Shahdezh mountain, 11 June 2010, Naderi & Jafari 1014 (HUI 22981)	OM141025	OM033049	OM048529	OM056568	
	(5)	Canada: Ontario, Sarnia, 15 June 1901, Macoun s.n. (CAN 38975)	OM141026	OM033050	OM048530	OM056569	
Bromus bromoideus (Lej.) Crép.	(1)	Belgium: 27 June 1909, Petry s.n. (W 1918-0004561)	OM141027	OM033051	OM048531	OM056570	
	(2)	Belgium: Alonso 349	KP987319 2	KP996932 2	×	×	
Bromus carinatus Hook. & Arn.		USA: California, Humboldt Co., 8.2 mi NW of Honeydew on Lost Coast Highway and 7.7 mi SE of Petrolia at bridge crossing Mattole River; along N edge of Mattole River, 6 August 2006, Peterson, Saarela & Sears 19700 (CAN 593840)	KM077294 1	KJ632444 1	×	×	
Canada: British Columbia, 4 mi SW of Hwy 37 on road towards New Aiyansh along Cranberry River, 21 July 2004, Peterson, Saarela & Smith 18689 (CAN 3524087)	×	×	MF597197 4	×	
Bromus catharticus Vahl	(1)	Iran: Isfahan, Faridan, 30 April 1998, without collector name (HUI 22978)	OM141028	OM033052	OM048532	OM056571	
	(2)	Spain: Acedo 23	KP987383 2	KP996906 2	×	×	
South Korea: Gwangyang-si, Jeollanam-do, 10 May 2012 (HCCN-PJ008548-PB-11)	×	×	KF713092 5	×	
Bromus commutatus Schrad.	(1)	Armenia: Syunik, 8 July 2003, Fayvush et al. 03-1334 (W 2005-0018638)	OM141029	OM033053	OM048533	OM056572	
	(2)	Austria: Wien, 13 June 2016, Adler s.n. (W 2016-0010156)	OM141030	OM033054	OM048534	OM056573	
	(3)	USA: North Carolina, Wake County, Mitchell Mill State Natural Area, SW side of Hwy 96 at the Little River, 23 May 2009, Rothfels, Cook & Pokorny 2860 (CAN 593689)	OM141031	OM033055	OM048535	OM056574	
Bromus danthoniae Trin.	(1)	Iran: Semnan, Shahmirzad, Chashm to Hiqu, 16 May 2016, Khazempour-osaloo et al. s.n. (TMUH 2016-001)	OM141032	OM033056	×	OM056575	
	(2)	Iran: Tehran, Damavand, Kilan, FiruzKuh road, 24 May 2016, Nasiri s.n. (TMUH 2016-002)	OM141033	×	×	OM056576	
	(3)	Iran: Markazi, Zarandieh, desert around Zarandieh, road highway Tehran to Qom, 25 April 2016, without collector (TMUH 2016-003)	OM141034	×	×	OM056577	
	(4)	Iran: Hamedan, after Dukhan toward Hamedan, 110 km Hamedan, 25 April 2016, without collector (TMUH 2016-004)	OM141035	×	×	OM056578	
Bromus cf. danthoniae	(5)	Iran: East Azerbaijan, inter Ahar and Hurand, 23 September 2007, Yousefi s.n. (HCAT Yousefi)	OM141036	OM033057	OM048536	OM056579	
	(6)	Iran: Yazd, Bafq, Gazestan village, 28 April 2011, Naderi s.n. (HUI 22971)	OM141037	OM033058	OM048537	OM056580	
Bromus danthoniae var. danthoniae	(1)	Iran: Chaharmahal and Bakhtiari, 55 km from Shahr-e Kord to Kouhrang, 24 May 2003, Aryavand s.n. (HUI 22964)	OM141038	OM033059	OM048538	OM056581	
	(2)	Iran: Lorestan, after Malayer toward Ilam, near wheat field, 27 May 2001, Aryavand s.n. (HUI 22965)	OM141039	OM033060	OM048539	OM056582	
	(3)	Iran: Yazd, Taft, after Sanij village, Shir kuh mountain, 16 May 2010, Naderi 1026 (HUI 22966)	OM141040	OM033061	OM048540	OM056583	
Bromus danthoniae var. pauciaristatus Naderi	(1)	Iran: Tehran, Haraz road, after Abali, 24 May 2016, Nasiri s.n. (TMUH 2016-005)	OM141041	×	×	OM056584	
	(2)	Iran: Kurdistan, Sanandaj, Zaleh station, 19 May 1986, Fattahi & Khaledian s.n. (TARI 1192)	OM141042	OM033062	OM048541	OM056585	
	(3)	Iran: Kohgiluyeh and Boyer-Ahmad, Sisakht, 20 April 2010, Naderi 1063 (HUI 22963)	OM141043	OM033063	OM048542	OM056586	
	(4)	Iran: Razavi Khorasan, Darrud to Jaghargh, Binalud mountain, 3 June 2010, Naderi & Hosseini 1065 (HUI 22967)	OM141044	OM033064	OM048543	OM056587	
	(5)	Iran: West Azerbaijan, 50 km to Sardasht from Mahabad, 28 May 1978, Runemark & Mozaffarian s.n. (TARI 29130)	OM141045	OM033065	OM048544	OM056588	
	(6)	Iran: Kurdistan, Baneh, 25 km from Baneh to Sanandaj, route of Nekerouz mountain, 9 May 2009, Hamzeh’ee et al. s.n. (TARI 95177)	OM141046	OM033066	OM048545	OM056589	
	(7)	Afghanistan: from seed, cultivated by R. Keane in Berkshire, UK, seed accession PI21998929, 29 August 2001, Keane RK018 (CAN 589848)	OM141047	OM033067	OM048546	OM056590	
Bromus densus Swallen		Mexico: Coahuila, Sierra do Zapalinamé, along trail from El Cuatro to El Penitente, 28 September 2007, Peterson, Saarela & Gómez Pérez 21128 (US-3554616, barcode 00967019)	OM141048	OM033068	OM048547	OM056591	
Bromus diandrus Roth		Morocco: Nador, Beni Chiker, 2010, Chrtek & Docˇkalová s.n. (LEB)	KM077297 1	KJ632447 1	×	×	
Canada: British Columbia, Vancouver Island, Island View Park, S of Hwy. 17 between Sidney and Victoria, along coast, 4 June 2007, Saarela, Percy & Chang 847 (CAN 590484)	×	×	MF597206 4	×	
Bromus erectus Huds.		Spain: Acedo et al. 224	KP987399 2	KP996885 2	×	×	
United Kingdom: Somerset, North (6), 31 August 2010 (NMW 5211)	×	×	JN895105 6	×	
Bromus gedrosianus Pénzes	(1)	Iran: Sistan and Baluchestan, Bampur road to Jolgeh-ye Chahhashem, 29 March 1988, without collector (HUI 22959)	OM141049	OM033069	OM048548	OM056592	
	(2)	Iran: Sistan and Baluchestan, Taftan, region Kharestan, 27 May 1985, Mozaffarian s.n. (TARI 53031)	OM141050	OM033070	OM048549	OM056593	
	(3)	Iran: Sistan and Baluchestan, Zabol, alfalfa field, 30 January 1965, Valizadeh & Ramak Maassoumi s.n. (TARI 36)	OM141051	OM033071	OM048550	OM056594	
Bromus gracillimus Bunge	(1)	Tajikistan: Pamir, 1958, Tolmatcheva s.n. (CAN)	KM077289 1	KJ632439 1	×	×	
	(2)	Iran: Kerman, Kuh-e Lalezar, 12 May 2011, Naderi s.n. (TARI 1126)	OM141052	OM033072	OM048551	OM056595	
Bromus grossus Desf. ex DC.		Austria: Wien, 27 June 1925, Seiller s.n. (W 2015-0007797)	OM141053	OM033073	OM048552	OM056596	
Bromus hordeaceus L.	(1)	Canada: Ontario, Niagara, Baden-Powell Park (city of Niagara Falls), 17 June 2008, Oldham et al. 35362 (CAN 605817)	OM141054	OM033074	OM048553	OM056597	
	(2)	Canada: British Columbia, roadside pullout on W side of Hwy. 1 (Trans-Canada) at Hells Gate, 22 June 2006, Saarela 734 (CAN 590431)	OM141055	OM033075	OM048554	OM056598	
	(3)	Canada: Ontario, Prince Edward County, Gull Bar, south shore of Prince Edward County, on Lake Ontario, 13 June 2008, Oldham & Brinker 35337 (CAN 605811)	OM141056	OM033076	OM048555	OM056599	
	(4)	Canada: British Columbia, Osoyoos, 0.5 km up Grizzly Road, S of Hwy. 3, ca 5 km E of Osoyoos, 20 June 2006, Saarela, Sears & Maze 646 (CAN 590394)	OM141057	OM033077	OM048556	OM056600	
	(5)	Canada: British Columbia, Kuskanook Rest Area, Kuskanook Harbour society, gravel around edge of lake and edge of mixed woods, 24 May 2006, Saarela 452 (CAN 590329)	OM141058	OM033078	OM048557	OM056601	
	(6)	Spain: Badajoz, Calera de León, 2011, Acedo et al. 109 (LEB)	KM077298 1	KJ632448 1	×	×	
Canada: British Columbia, Vancouver Island, Thetis Lake Regional Park, just N of Hwy. 1, N of Langford, 4 June 2007, Saarela, Percy & Chang 0859 (CAN 590505)	×	×	MF597209 4	×	
Bromus induratus Hausskn. & Bornm.		Iran: East Azerbaijan, Marand, ca. 35 km N of Marand, Kiamaki-Dagh mountain, 25 July 1990, Assadi & Olfat s.n. (TARI 68598)	OM141059	OM033079	OM048558	OM056602	
Bromus inermis Leyss.		Russia: Buryatiya Republic, Kyakhtinskii Raion, 2010, Chepinoga 28355 (LEB)	KM077290 1	KJ632440 1	×	×	
Mongolia: grown from seed accession W6 21403, USDA Nat. Small Grain Coll., Aberdeen, Idaho, USA	×	×	KY636082 7	KY636082 7	
Bromus intermedius Guss.	(1)	Turkey: Donmez 3350	KP987346 2	KP996958 2	×	×	
	(2)	Iran: Gilan, Manjil, 22 May 1973, Sabeti s.n. (TARI 10565)	×	OM033080	×	OM056603	
Bromus interruptus (Hack.) Druce	(1)	Canada: Nova Scotia, Dalhousie University, grown in greenhouse, seed from K [Kew Gardens], 6 May 1985, Harvey s.n. (CAN 605189)	OM141060	×	OM048559	OM056604	
	(2)	United Kingdom: England	KP987347 2	KP996959 2	×	×	
United Kingdom: Chase 20653 (K)	×	×	GQ248089 8	×	
Bromus japonicus Thunb.	(1)	Iran: Ardabil, Sabalan mountain, 5 August 2009, Naderi 1017 (HUI 22970)	OM141061	OM033081	OM048560	OM056605	
	(2)	Iran: Kohgiluyeh and Boyer-Ahmad, the beginning of Yasuj road to Eqlid, before Chenar spring, in the forest of margin river, 5 May 2011, Naderi 1018 (HUI 22972)	OM141062	OM033082	OM048561	OM056606	
	(3)	Canada: Ontario, Prince Edward County, Miller tract, Hastings-Prince Edward Land Trust, hill top road, 20 June 2013, Oldham et al. 40762a (CAN 606737)	OM141063	OM033083	OM048562	OM056607	
	(4)	Canada: Ontario, Thunder Bay, S side of railway tracks near Neys Provincial Park, E of park entrance road, 4 km WNW of Coldwell, 28 June 2009, Oldham & Brinker 36166 (CAN 605819)	OM141064	OM033084	OM048563	OM056608	
	(5)	Canada: Ontario, Peterborough, Water Street at Parkhill Road, 2 June 2010, Oldham & Bowles 37317 (CAN 597881)	OM141065	OM033085	OM048564	OM056609	
Bromus kalmii A.Gray		USA: accession of unknown origin acquired in Everwilde Farm, 2013, Acedo 332 (LEB)	KM077292 1	KJ632442 1	×	×	
Canada: Quebec, Montreal, Island of Montreal, Jardin Botanique de Montreal, jardin alpin, 08 June 2012, Lambert s.n. (MT00179369)	×	×	MG216969 9	×	
Bromus kopetdagensis Drobow		Iran: Tehran, Road of Firuzkuh, Rostam Abad, 27 July 1972, Dini & Arazm s.n. (TARI 10649)	OM141066	OM033086	OM048565	OM056610	
Bromus lanceolatus Roth	(1)	Iran: Zanjan, Gilvān, after Manjil, 23 May 1973, Sabeti s.n. (TARI 10545)	×	×	×	OM056611	
	(2)	Iran: Khuzestan, Haft Tappeh, train station, 17 March 1986, Mozaffarian s.n. (TARI 62800)	×	×	×	OM056612	
	(3)	Iran: Khuzestan, 15 km to Shush from Dezful, 16 April 2010, Naderi 1021 (HUI 22954)	OM141067	OM033087	OM048566	OM056613	
	(4)	Iran: Fars, Kazerun, Famur Rural District, Qaleh-ye Narenji, 7 March 2010, Naderi 1132 (HUI 22955)	OM141068	OM033088	OM048567	OM056614	
Bromus macrocladus Boiss.		Iran: West Azerbaijan, ca. 70 km W Khoy, upper mountains of Razi village, 26 July 1990, Assadi & Olfat s.n. (TARI 68901)	OM141069	OM033089	OM048568	OM056615	
Bromus madritensis L.		Iran: Tehran, 84 km from Tehran to Qom, 18 May 1974, Amin & Bazargan s.n. (TARI 18217)	OM141070	OM033090	OM048569	OM056616	
Bromus oxyodon Schrenk	(1)	Iran: Razavi Khorasan, between Quchan and Dargaz, Tandooreh National Park, 28 May 1984, Assadi & Maassoumi s.n. (TARI 50702)	OM141071	OM033091	×	OM056617	
	(2)	Iran: Razavi Khorasan, Darrud to Jaghargh, Binalud mountain, 3 June 2010, Naderi & Hossaini 1020 (HUI 22957)	OM141072	OM033092	OM048570	OM056618	
	(3)	Iran: Razavi Khorasan, ca. 45 N. of Shirvan, Sarany protected area (EG3), 26 May 1984, Assadi & Maassoumi s.n. (TARI 50508)	OM141073	OM033093	OM048571	OM056619	
Bromus pannonicus subsp. monocladus (Domin) P.M.Sm.		Slovakia: Zilinský, 8 June 1994, Mikoláš 9157 (W 2013-0008548)	OM141074	OM033094	OM048572	OM056620	
Bromus pectinatus Thunb.	(1)	Belgium: grown from seed accession PI 442453, Keane 23 (ALTA-VP 110299)	AY367939 10	×	×	×	
	(2)	South Africa: Aizpuru et al. (LM6093)	KP987364 2	KP996977 2	×	×	
Bromus pseudobrachystachys H.Scholz	(1)	Iran: Mazandaran, Nur Forest Park, 9 July 2002, Sahebi s.n. (HUI 13901)	OM141075	OM033095	OM048573	OM056621	
	(2)	Iran: Fars, Kazerun, Qaleh-ye Narenji, Narges-Zar, 6 May 2011, Naderi 1001 (HUI 22962)	OM141076	OM033096	OM048574	OM056622	
Bromus pulchellus Fig. & De Not.	(1)	Iran: Sistan and Baluchestan, Zahedan, 30 km Zahedan to Zabol, 4 April 1983, Mozaffarian s.n. (TARI 42671)	OM141077	OM033097	OM048575	OM056623	
	(2)	Iran: Isfahan, side road highway Isfahan - Kashan, 25 km Abyaneh village, 5 May 2009, Naderi & Zoghi 1019 (HUI 22960)	OM141078	OM033098	OM048576	OM056624	
	(3)	Iran: Sistan and Baluchestan, between Iranshahr and Bam, Bazman, Kuh Khezr, 3 May 1977, Assadi s.n. (TARI 23179)	OM141079	OM033099	OM048577	OM056625	
Bromus pumilio (Trin.) P.M.Sm.	(1)	Iran: Hormozgan, Bandar Abbas, S side of kuh-e Geno, 5 April 1975, Wendelbo & Foroughi s.n. (TARI 15526)	OM141080	OM033100	OM048578	OM056626	
	(2)	Armenia: Nersesyan 50-2004	KP987312 2	KP996869 2	×	×	
Iran: Gilan (HAL 22065)	×	×	FM253120 11	×	
Bromus pumpellianus Scribn.		Russia: Magadanskaya Province, Chukotka Autonomous Region, Bilibinskiy Area, Anyuy Mountains, upper course of Pogynden River, on floodplain of Yagodniy creek, 15 July 1974, Koroleva & Bryzgalova s.n. (CAN 528978)	OM141081	OM033101	OM048579	OM056627	
Bromus racemosus L.	(1)	Iran: Gilan, 25 km to Masouleh from Fuman, 6 July 1995, Asadi s.n. (TARI 73729)	OM141082	OM033102	×	OM056628	
	(2)	Iran: Mazandaran, 33 km from Sari to Kiasar, Alamdar Deh, Doseleh, along rice field, 27 April 2009, Naderi 1002 (HUI 22969)	OM141083	OM033103	OM048580	OM056629	
	(3)	Canada: Ontario, Prince Edward County, Miller Tract, Hasting-Prince Edward Land Trust, Hill Top Road, 20 June 2013, Oldham et al. 40762b (CAN 606725)	OM141084	OM033104	OM048581	OM056630	
	(4)	Canada: Ontario, Niagara, ca 2.5 km west of Welland Canal at Port Colborne, just north of CN railway line, at Humberstone–Wainfleet Twp. boundary, 31 May 2006, Oldham 32598 (CAN 605805)	OM141085	OM033105	OM048582	OM056631	
	(5)	Canada: British Columbia, 10 km W of Princeton on Hwy. 3, 19 June 2006, Saarela, Sears & Maze 586A (CAN 590343)	OM141086	OM033106	OM048583	OM056632	
	(6)	Canada: British Columbia, Rocky Mountain Forest District, along Hwy. 3 1.5 E of Elko, 23 May 2006, Saarela 405A (CAN 590308)	OM141087	OM033107	OM048584	OM056633	
	(7)	Canada: British Columbia, Osoyoos, Haynes Provincial Park, 25 May 2006, Saarela 484 (CAN 591393)	OM141088	OM033108	OM048585	OM056634	
	(8)	Denmark: Jutland, 28 June 1969, Jacobsen & Svendsen s.n. (CAN 344128)	OM141089	OM033109	OM048586	OM056635	
Bromus ramosus Huds.		Bulgaria: Aedo et al.	KP987418 2	KP996925 2	×	×	
Besnard 342004 (G)	×	×	HE586076 12	×	
Bromus rechingeri Melderis		Iran: Zahedan, 18 km SE Zahedan, 9 April 1969, Babakhanlou s.n. (TARI 9465)	OM141090	OM033110	OM048587	OM056636	
Bromus rigidus Roth		Iran: Alonso 347	KP987437 2	KP996887 2	×	×	
South Korea: Wando, Jeollanam-do, 28 May 2013 (HCCN-PJ008548-PB-383)	×	×	KF713103 13	×	
Bromus rubens L.		Iran: Kohgiluyeh and Boyer-Ahmad, Gachsaran, 35 km Gachsaran to Noor Abad, 17 April 2010, Naderi 1116 (HUI 22947)	OM141091	OM033111	OM048588	OM056637	
Bromus sclerophyllus Boiss.		Turkey: Antalya, 18 June 1992, Parolly,G. A 9-3 (W 2015-0009990)	OM141092	OM033112	OM048589	OM056638	
Bromus scoparius L.	(1)	Iran: Khuzestan, near to Shatt-e Izeh (wetland), 15 April 2010, Naderi s.n. (TARI 1011)	OM141093	OM033113	OM048590	OM056639	
	(2)	Iran: Chaharmahal and Bakhtiari, Nghan, road Naghan to Izeh, river side, 14 April 2010, Naderi 1006-1 (HUI 22958)	OM141094	OM033114	OM048591	OM056640	
	(3)	Iran: West Azerbaijan, SW of Rezaiyeh (Urmia), Silvana valley, along the road SE of Dizeh, 25 May 1976, Runemark & Forughi s.n. (TARI 19846)	OM141095	OM033115	OM048592	OM056641	
	(4)	Iran: Kermanshah, Sarpol-e Zahab, Sarab-e Garm village, 19 April 1989, Hatami s.n. (TARI 2312)	OM141096	OM033116	OM048593	OM056642	
	(5)	Iran: Mazandaran, W of Ramsar, W of Javaher Deh, 28 June 1976, Runemark & Ramak Maassoumi s.n. (TARI 20799)	OM141097	OM033117	OM048594	OM056643	
Bromus cf. scoparius	(6)	Greece: Kérkyra (Island), 9 May 2000, Gutermann et al. 34657 (W 2006-0015232)	OM141098	OM033118	OM048595	OM056644	
Bromus secalinus L.	(1)	Russia: Leningrad (St. Petersburg), Lyzhskii area, near village Kolentsevo, near rye field, 22 July 1964, Ber 149 (CAN 327328)	OM141099	OM033119	OM048596	OM056645	
	(2)	Sweden: Dalarna, 23 July 1919, Samuelsson 260 (CAN 132696)	OM141100	OM033120	OM048597	OM056646	
	(3)	Finland: Al saitvin, s.d., Haeggstrom 9920 (H 836530)	OM141101	OM033121	OM048598	OM056647	
	(4)	Germany: accession 07-104-07-74 Botanischer Garten Oldenburg, 2011, Alonso 339 (LEB)	KM077304 1	KJ632454 1	×	×	
United Kingdom: Cardiganshire, 8 October 2000, Chater 00/397 (NMW.V.2003.14.92)	×	×	JN895850 6	×	
Bromus sewerzowii Regel	(1)	Afghanistan: 7 May 1967, Rechinger 34038 (W 1968-0006905)	OM141102	OM033122	OM048599	OM056648	
	(2)	Kazakhstan: Zhambyl, 4 June 1967, Demina, O. Gerbarii Flory SSSR 4930 (W 1972-0018335)	OM141103	OM033123	OM048600	OM056649	
Bromus squarrosus L.	(1)	Armenia: Vayots Dzor, 18 June 2004, Fayvush et al. 04-0692 (W 2006-0004999)	OM141104	OM033124	OM048601	OM056650	
	(2)	Georgia: Tbilisi, 21 May 2005, Lachashvili s.n. (W 2007-0004141)	×	×	OM048602	OM056651	
	(3)	Greece: Arkodia, 19 May 1986, Burri & Krendl s.n. (W 2008-0010938)	OM141105	OM033125	OM048603	OM056652	
	(4)	Austria: Wien, 8 June 2017, Adler s.n. (W 2017-0010247)	OM141106	OM033126	OM048604	OM056653	
	(5)	Canada: Ontario, Prince Edward County, Ameliasburgh Township, Massasauga Point, ca. 5 km SE of Belleville, 23 June 1996, Oldham & Blaney 18625 (CAN 609782)	OM141107	OM033127	OM048605	OM056654	
	(6)	Spain: León, Llombera, 2010, Llamas & Acedo 43 (LEB)	KM077303 1	KJ632453 1	×	×	
Canada: British Columbia, W of Osoyoos, 1.3 km up Richter Mountain Road, 19 June 2006, Saarela, Sears & Maze 628 (CAN 590361)	×	×	MF597248 4	×	
Bromus sterilis L.		Iran: Isfahan, road of Semirom to Yasuj, Ab-e-garm gah, near to Rood Abad village, near river, 29 April 2009, Naderi & Zoghi 1146 (HUI 22949)	OM141108	OM033128	OM048606	OM056655	
Bromus tectorum L. subsp. tectorum		Iran: Fars, Dasht-e Arzhan, 18 April 2010, Naderi 1100 (HUI 22952)	OM141109	OM033129	OM048607	OM056656	
Bromus tomentellus Boiss.		Iran: Fars, Dasht-e Arzhan, 18 April 2010, Naderi 1100 (HUI 22974)	OM141110	OM033130	OM048608	OM056657	
Bromus tomentosus Trin.	(1)	Iran: Tehran, Gajereh, road of Karaj-Chalus, 6 August 1972, Babakhanlou & Amin s.n. (TARI 10553)	OM141111	OM033131	OM048609	OM056658	
	(2)	Iran: Mazandaran, Pol-e Zangoleh to Nasan, before Golestanak area, 12 August 2009, Naderi 1121 (HUI 22975)	OM141112	OM033132	OM048610	OM056659	
Bromus vulgaris (Hook.) Shear		USA: California, 10.2 mi NW of Philo on Hwy. 128 towards Albion at Navarro Redwoods State Park, 5 August 2006, Peterson et al. 19695 (CAN-593921)	KX872936 14	KX872308 14	×	×	
Canada: British Columbia, Mayne Island, Bennett Bay, Gulf Islands National Park Reserve of Canada, Wilkes Road off Bennett Bay Road, 3 June 2007, Saarela, Percy & Chang 822 (CAN 590469)	×	×	KM974737 15	KM974737 15	
Hordeum marinum Huds.		Spain: Zamora, Villafáfila, 2012, Acedo & Llamas 263 (LEB)	KM077287 1	KJ632437 1	×	×	
Spain: Toledo, Jacobsen s.n., accession BCC 2006-	KU513491 16	×	KY636106 7	KY636106 7	
Hordeum vulgare L.		Iran: Isfahan, 35 km SW of Natanz, Yahya Abad village, 27 May 2010, Abbasi & Afsharzadeh s.n. (HUI 17603)	OM141113	×	OM048611	OM056660	
Littledalea alaica (Korsh.) Petrov ex Kom.		Tajikistan: Gorno-Dabakhshankaya Autonomous Region, Kainda River, Tzelev 1335 (LE)	FM179415 11	×	×	×	
China: voucher deposited at School of Life Science, Qinghai Normal University	×	×	MG570144 17	MG570144 17	
Littledalea racemosa Keng		China: voucher L122	×	×	MF614917 18	MF614917 18	
Triticum aestivum L.		GZ168 cultivar	×	×	KJ592713 19	KJ592713 19	
Turkey	AY450258 20	×	×	×	
Triticum turgidum L.		Spain: Llamas et al. 95.2010 (LEB)	KP296130 21	KP325375 21		×	
cultivar TA2836	×	×	KJ614397 22	KJ614397 22	
Notes:

× = not available. Superscripts identify the authors of previously published or unpublished sequences.

1 Alonso et al. (2014).

2 A. Alonso, F. Llamas, M. Pimentel, C. Acedo, unpublished data, 2016.

3 Seberg & Petersen (2007).

4 D. M. Percy, unpublished data, 2018.

5 J. Lee, C-S. Kim, I-Y. Lee, unpublished data, 2014.

6 de Vere et al. (2012).

7 Bernhardt et al. (2017).

8 CBOL Plant Working Group et al. (2009).

9 Kuzmina et al. (2017).

10 Saarela et al. (2007).

11 Schneider et al. (2009).

12 Grass Phylogeny Working Group II (2012).

13 J. Lee, C-S. Kim, I-Y. Lee, unpublished data, 2012.

14 Saarela et al. (2017).

15 Saarela et al. (2015).

16 Mahelka et al. (2017).

17 X. Su, unpublished data, 2018.

18 Liu et al. (2017).

19 Bahieldin et al. (2014).

20 Gulbitti-Onarici et al. (2009).

21 C. Acedo, A. Alonso, F. Llamas, unpublished data, 2018.

22 Gornicki et al. (2014).

DNA extraction protocol

We extracted DNA at Tarbiat Modares University and the Canadian Museum of Nature. At Tarbiat Modares University, genomic DNA was extracted from fresh or dried material, mostly using Plant DNA kits (Exgene TM Plant SV mini, GeneAll Biotechnology Co., Seoul, South Korea) following the manufacturer’s instructions or using a modified cetyltrimethylammonium bromide (CTAB) method (Doyle & Doyle, 1987). At the Canadian Museum of Nature, we extracted DNA from both silica-gel-preserved leaf and herbarium samples following a silica-membrane column purification protocol similar to commercially available DNA extraction kits (modified from Alexander et al., 2007). The success of DNA extractions was assessed via gel electrophoresis in 1.25% agarose gels stained with ethidium bromide.

Amplification and sequencing

We amplified two nrDNA regions (ITS, including ITS1, 5.8S, ITS2, and partial 26S, and ETS, including ETS1, and ETS1f) and two plastid regions (rpl32-trnLUAG and matK). We amplified ITS, ETS, and matK at Tarbiat Modares University using the following primer pairs and reaction conditions: ITS—primers ITS5m (Sang, Crawford & Stuessy, 1995) and ITS4 (White et al., 1990) at 95 °C for 4 min, 30 cycles of 95 °C for 1 min, 53.5 °C for 40 s, 72 °C for 1 min and a final extension step of 72 °C for 6 min; ETS—primers RETS-B4F and RETS-B3F (Alonso et al., 2014) and 18S-R (Starr, Harris & Simpson, 2003) at 95 °C for 2 min, 29 cycles of 95 °C for 45 s, 58 °C for 45 s, 72 °C for 2 min and a final extension step of 72 °C for 5 min; matK—primers matK-AF and matK-BF (Ooi et al., 1995) and trnK-2R (Steele & Vilgalys, 1994) at 95 °C for 1 min, 35 cycles of 95 °C for 30 s, 55 °C for 40 s, 72 °C for 1 min and a final extension step of 72 °C for 10 min. Amplification reactions were performed in 20 μL volumes containing 8 μL deionized water, 10 μL of Taq DNA Polymerase 2x Master Mix RED (Ampliqon, Denmark, Copenhagen), 0.5 mL of each primer (10 pmol/mL), and 1 μL of template DNA (ca. 20 ng/mL). The quality of amplification products was checked via gel electrophoresis in 1% agarose gels stained with ethidium bromide. Amplification products and primers used for amplification were sent to Pishgam Biotech Co. (Tehran, Iran) for Sanger sequencing by Macrogen (Seoul, South Korea) using an Applied Biosystems Prism 3730xl DNA Analyzer (Thermo Fisher Scientific, Waltham, MA, USA).

We amplified some samples at the Canadian Museum of Nature. For ITS, ETS, and matK, for most samples we used the same primers used at Tarbiat Modares University. For some older herbarium samples, however, we used alternate primers. For ITS these were KRC (Torrecilla & Catalán, 2002), ITS-p5 (Cheng et al., 2016), and 26SE (Sun et al., 1994). For matK these were matK_ag520F, matK_po3R, matK_po1F (Saarela et al., 2017) and a new internal primer we designed: matK_ag653R (5′-TTAGATGGAYCCTTCGCGGC-3′). We amplified the rpl32-trnLUAG region at the Canadian Museum of Nature using primers rpl32-F and trnL(UAG) (Shaw et al., 2007). We used two DNA polymerases and thermal cycling programs for amplifying the four DNA regions: (1) 15 μL volume with 8.6 μL of ddH20, 3 μL of 5X reaction buffer, 0.3 μL of 10 mM dNTP, 0.75 μL of 10 μM primer, 0.45 μL of dimethyl sulfoxide, 0.3 U of Phusion DNA Polymerase (New England BioLabs Inc., Ipswich, MA, USA), and 1 μL of DNA template (1:10 dilution) at 98 °C for 30 s, 34 cycles of 98 °C for 10 s, 56 °C for 30 s, 72 °C for 30 s, and a final extension step of 72 °C for 5 min; (2) 15 μL volume with 11.3 μL of ddH20, 1.5 μL of 10X reaction buffer, 0.3 μL of 10 mM dNTP, 0.375 μL of 10 μM each primer, 0.75 U of DreamTaq DNA Polymerase (Thermo Fisher Scientific, Waltham, MA, USA), and 1 μL of DNA template (1:10 dilution) at 95 °C for 3 min, 35 cycles of 95 °C for 30 s, 55 °C for 30 s, 72 °C for 1.5 min, and a final extension step of 72 °C for 10 min. Amplification success was assessed via gel electrophoresis in 1.25% agarose gels stained with ethidium bromide.

Sequencing reactions at the Canadian Museum of Nature were performed in 10 μL reactions containing 6.2 μL of DNA-grade H2O, 1.8 μL of 5X reaction buffer, 0.5 μL of primer, 0.5 μL of BigDye Terminator v3.1 Ready Reaction Mix (Thermo Fisher Scientific, Waltham, MA, USA), and 1 μL of diluted PCR product. The reaction program consisted of 95 °C for 3 min, 30 cycles of 96 °C for 30 s, 50 °C for 20 s, and 60 °C for 4 min. We purified reaction products via an EDTA-NaOH-ethanol precipitation protocol recommended by the sequencing kit manufacturer. Purified DNA pellets were resuspended in Hi-Di Formamide, denatured at 95 °C for 5 min, cooled for 2 min, and sequenced via automated capillary electrophoresis on an Applied Biosystems 3500xL Genetic Analyzer (Thermo Fisher Scientific, Waltham, MA, USA).

Data assembly and phylogenetic analysis

Sequences were assembled, trimmed, and visually assessed using Geneious 11.1.5 software (https://www.geneious.com, Kearse et al., 2012). Therein, we replaced suspect base codes with nucleotide ambiguity codes, we confirmed open reading frames in protein-coding genes by screening for stop codons, and we aligned edited sequences for each gene separately using the MAFFT (Multiple Alignment using Fast Fourier Transform) v7.388 alignment algorithm with default settings (Katoh & Standley, 2013). Four matrices (one per region) of aligned sequences were individually exported from Geneious as NEXUS files, then concatenated as two combined nuclear and plastid DNA matrices using SequenceMatrix (Vaidya, Lohman & Meier, 2011). We saved these matrices as NEXUS files (Supplemental Information, Datasets S1 and S2).

We conducted phylogenetic analyses for each DNA region (single-region analysis) and for the two combined datasets (nuclear, plastid). Each region and each codon position, in the case of protein-coding genes, was treated as a separate partition. For rpl32-trnLUAG, which includes a protein-coding gene and two non-coding regions (the intergenic spacer and part of trnLUAG), we defined each codon position and the two non-coding regions as separate data blocks. Incongruence between the combined datasets was statistically evaluated using the incongruence length difference (ILD) test (Farris et al., 1994), implemented as the partition homogeneity test in PAUP* v4.0a168 (Swofford, 2002) with 1,000 replicates, simple addition of taxa, tree-bisection-reconnection branch swapping, multitrees in effect, and saving 10 trees per replicate.

We performed maximum parsimony (MP) analyses in PAUP* for the combined nuclear and chloroplast DNA datasets, with 1,000 random replications in the heuristic searches, using tree bisection-reconnection branch swapping and 100 random addition sequence replicates. Branch support values were estimated using full heuristic searches with 1,000 bootstrap replicates, each with simple sequence addition and one tree held per replicate. We performed maximum likelihood (ML) analyses using W-IQ-TREE (Trifinopoulos et al., 2016), available at http://iqtree.cibiv.univie.ac.at. The models for ML analyses were selected using Model Finder (Chernomor, von Haeseler & Minh, 2016; Kalyaanamoorthy et al., 2017) implemented in IQ-TREE web; this analysis identified models SYM+R3 and GTR+F+R2 as best fit for combined nuclear and combined plastid DNA, respectively. Bootstrap support for ML trees was determined based on 1,000 ultrafast bootstrap replicates with one search replicate per bootstrap replicate (Minh, Nguyen & von Haeseler, 2013) and the other options in the default setting.

Bayesian inference (BI) analyses for single regions and the two combined datasets were performed in MrBayes v.3.2.7a (Ronquist et al., 2012) at the CIPRES Science Gateway V. 3.3 (http://www.phylo.org/; Miller, Pfeiffer & Schwartz, 2010). We used PartitionFinder2 (Lanfear et al., 2016) to determine the partitioning scheme and best-fit models of molecular evolution for each gene region and the two concatenated alignments using the corrected Akaike information criterion (AICc). The best models were SYM+I+G and GTR+I+G for nuclear and plastid DNA sequences, respectively. The best substitution models for each subset of sites are presented in Supplemental Information (Datasets S3 and S4). For each BI analysis, we ran two parallel runs with four Markov chain Monte Carlo (MCMC) heuristic searches per run for 5 ×107 generations, sampling the chains every 1,000th generation. We discarded the first 25% of trees from each run as burn-in and stopped each analysis after the standard deviation of split frequencies dropped below 0.01. The resulting trace files were checked using Tracer v.1.7.1 (Rambaut et al., 2018) to ensure that effective sample size values were >200 for all parameters. All trees were visualized and partially edited in FigTree v.1.4.4 (Rambaut, 2018).

Results

Phylogenetic analyses

We produced 383 new sequences from 104 samples: 96 ITS, 93 ETS, 103 rpl32-trnLUAG, and 91 matK. ITS and ETS alignments consisted of 928 sites for 124 taxa and 1,139 sites for 116 taxa, respectively, of which 214 (23.1%) and 398 (34.9%) sites were variable and 177 (19.1%) and 284 (24.9%) sites were potentially parsimony informative. The mean G + C content of ITS and ETS regions was 56.6% and 53.5%, respectively. The BI cladograms obtained in nuclear ITS and ETS analyses are presented under Supplemental Information (Figs. S1 and S2) along with the bootstrap support values from the ML analyses.

The aligned rpl32-trnLUAG and matK sequences produced a matrix of 1,065 sites for 110 taxa and 1,443 sites for 111 taxa, respectively. For rpl32-trnLUAG, 127 (11.9%) sites were variable and 95 (8.9%) were potentially parsimony informative. For matK, 133 (9.2%) sites were variable and 103 (7.1%) were potentially parsimony informative. The mean G + C content of the rpl32-trnLUAG and matK regions was 26.4% and 33.3%, respectively. The BI cladograms, along with the bootstrap support values from the ML analyses, for rpl32-trnLUAG and matK data are presented in the Supplemental Information (Figs. S3 and S4).

The nrDNA data resolves relationships among Bromus sections better than the plastid data, and relationships among most species of B. sect. Bromus are unresolved in both nuclear and plastid trees. All plastid trees have higher consistency indices (CI) and retention indices (RI) compared with nrDNA trees. These two indices are also higher in the combined plastid data (CI = 0.792, RI = 0.950) than the combined nuclear data (CI = 0.608, RI = 0.900). These data indicate more homoplasy in the nuclear data than in the plastid data.

The phylogenies derived from the combined nuclear data and the combined plastid data were topologically incongruent, and the ILD test for these two data partitions had a P value of <0.01. Therefore, we did not combine the nuclear and plastid datasets. We interpreted MP bootstrap support (PB) and ML bootstrap support (LB) values of 90–100% as strong support, 70–89% as moderate, and 50–69% as weak, and we interpreted Bayesian posterior probabilities (PP) ≥0.95 as strong support.

Analysis of combined nuclear ribosomal sequences

The combined nuclear dataset was 2,067 bp and included 125 taxa, with 1,606 potentially parsimony informative characters and a mean G + C content of 55.1%. Maximum parsimony, maximum likelihood, and Bayesian inference analyses of the combined nuclear matrix produced similar topologies, but the BI tree is better resolved and supported than the other trees. The BI tree, including the support values from the ML and MP analyses for shared clades, is shown in Figs. 1 and 2. Individual ML and MP trees are presented in Supplemental Information (Figs. S5 and S6).

Figure 1 A portion of the majority rule consensus tree inferred from Bayesian analysis of nrDNA ITS + ETS sequences.

Bayesian posterior probabilities, maximum likelihood bootstrap support, and maximum parsimony bootstrap support are indicated above the branches, respectively. Posterior probabilities <0.5 and bootstrap support <50% are indicated with a hyphen.

Figure 2 A portion of the majority rule consensus tree inferred from Bayesian analysis of nrDNA ITS + ETS sequences.

Bayesian posterior probabilities, maximum likelihood bootstrap support, and maximum parsimony bootstrap support are indicated above the branches, respectively. Posterior probabilities <0.5 and bootstrap support <50% are indicated with a hyphen.

In the phylogenetic trees derived from nrDNA sequences, Bromus s.l. is monophyletic with strong support (PP = 1, LB = 100%, PB = 98%). The genus is divided into two major clades. One clade is weakly supported (PP = 0.83, LB = 85%, PB = 67%) and comprises B. densus (B. sect. Mexibromus), B. pumilio (B. sect. Boissiera), and B. gracillimus (B. sect. Nevskiella). Bromus pumilio and B. gracillimus are sister taxa (PP = 1, LB = 100%, PB = 100%). The other clade is strongly supported (PP = 1, LB = 100%, PB = 97%) and comprises two subclades. One subclade comprises all species of B. sects. Bromopsis, Ceratochloa, Genea, and Neobromus and one species of B. sect. Bromus (B. rechingeri, of the B. pectinatus complex). Bromus section Bromopsis is not monophyletic. Within this subclade, B. ramosus Huds. and B. vulgaris (Hook.) Shear–B. kalmii A.Gray form a weakly to strongly supported clade (PP = 0.97, LB = 97%, PB = 70%) sister to a maximally supported clade comprising the remaining species. Within the latter clade, B. sect. Ceratochloa species form a strongly supported subclade (PP = 1, LB = 99%, PB = 98%) sister to B. berteroanus (B. sect. Neobromus) (PP = 0.96, LB = 96%, PB = 69%). Bromus rechingeri and all B. sect. Genea species form a strongly supported clade (PP = 1, LB = 99%, PB = 93%). Within this clade, B. rechingeri, B. sterilis, and B. diandrus–B. rigidus Roth form a maximally supported subclade, but relationships among these three lineages are unresolved. Bromus tomentellus (B. sect. Bromopsis) is weakly to strongly supported (PP = 1, LB = 95%, PB = 65%) as sister to the B. sect. Genea–B. rechingeri clade, and these lineages are part of a broader weakly to strongly supported clade including five other B. sect. Bromopsis species (B. erectus, B. pannonicus subsp. monocladus, B. kopetdagensis, B. induratus, and B. sclerophyllus).

The second subclade (Bromus sect. Bromus clade) is weakly to moderately supported (PP = 0.84, LB = 89%, PB = 65%). It comprises all B. sect. Bromus species except B. rechingeri. Within this clade, B. alopecuros subsp. caroli-henrici is sister to the remaining samples of the clade, which form a maximally supported clade, Bromus sect. Bromus clade A. Within Bromus sect. Bromus clade A, B. hordeaceus and B. interruptus form a strongly supported clade (PP = 1, LB = 100%, PB = 99%) sister to a strongly supported clade, Bromus sect. Bromus clade B (PP = 1, LB = 100%, PB = 99%), comprising the remaining species. Within Bromus sect. Bromus clade B, a maximally supported subclade comprising B. scoparius is sister to a moderately to strongly supported clade, Bromus sect. Bromus clade C. This clade includes several sublineages; relationships among them are poorly resolved. Bromus briziformis, B. sewerzowii, and B. danthoniae (B. danthoniae var. danthoniae and B. danthoniae var. pauciaristatus) form a strongly supported subclade (PP = 1, LB = 98%, PB = 78%). The three B. gedrosianus samples form a moderately to strongly supported subclade (PP = 0.99, LB = 82%, PB = 98%). The three B. oxyodon samples form a weakly to strongly supported subclade (PP = 0.97, LB = 81%, PB < 50%). The three B. pulchellus samples form a moderately to strongly supported subclade (PP = 1, LB = 100%, PB = 84%). Bromus oxyodon and B. pulchellus from a weakly to strongly supported clade (PP = 0.98, LB = 76%). One sample of B. bromoideus is weakly to strongly supported as sister to the aforementioned lineages.

Of the remaining Bromus sect. Bromus clade C species, a subset form several subclades: (i) a weakly to strongly supported subclade (PP = 0.87, LB = 97%) comprising two samples of B. intermedius Guss., three of B. japonicus, one of B. racemosus, two of B. lanceolatus, one of B. squarrosus, two of B. pectinatus, and B. arenarius; (ii) a strongly supported subclade (PP = 1, LB = 99%, PB = 68%) comprising three samples of B. arvensis, one of B. grossus, and one of B. racemosus; (iii) a weakly to strongly supported subclade (PP = 0.99, LB = 100%, PB = 53%) comprising one B. commutatus sample and two each of B. japonicus and B. racemosus; (iv) a weakly to strongly supported subclade (PP = 0.76, LB = 93%) comprising two B. secalinus samples. Multiple individuals of B. racemosus, B. secalinus, B. squarrosus, and B. pseudobrachystachys, one individual of B. arvensis, one of B. bromoideus, and the one sampled individual of B. macrocladus fall along the Bromus sect. Bromus clade C backbone.

Analysis of combined plastid sequences

The combined matrix of plastid rpl32-trnLUAG and matK had 2,508 aligned positions and 123 taxa, with a mean G + C content of 30.3%. The BI tree derived from the combined plastid matrix, including the support values from the ML and MP analyses, is shown in Figs. 3 and 4. Individual ML and MP trees are presented in Supplemental Information (Figs. S7 and S8). Bromus is maximally supported (PP = 1, LB = 100%, PB = 100%) as monophyletic. The genus is divided into two major clades: (i) a strongly supported clade (PP = 1, LB = 100%, PB = 99%) comprising sister groups B. gracillimus and B. pumilio (PP = 0.99, LB = 95%, PB = 81%) and a moderately to strongly supported subclade (PP = 1, LB = 100%, PB = 80%) corresponding to B. sect. Ceratochloa; (ii) a weakly to moderately supported clade (PP = 0.84, LB = 73%, PB = 62%) comprising B. densus (B. sect. Mexibromus) and a maximally supported subclade including all species of B. sects. Bromopsis, Genea, and Bromus.

Figure 3 A portion of the majority rule consensus tree inferred from Bayesian analysis of cpDNA rpl32-trnLUAG + matK sequences.

Bayesian posterior probabilities, maximum likelihood bootstrap support, and maximum parsimony bootstrap support are indicated above the branches, respectively. Posterior probabilities <0.5 and bootstrap support <50% are indicated with a hyphen.

Figure 4 A portion of the majority rule consensus tree inferred from Bayesian analysis of cpDNA rpl32-trnLUAG + matK sequences.

Bayesian posterior probabilities, maximum likelihood bootstrap support, and maximum parsimony bootstrap support are indicated above the branches, respectively. Posterior probabilities <0.5 and bootstrap support <50% are indicated with a hyphen.

Bromus sect. Bromopsis species except B. tomentosus form a weakly to strongly supported subclade (PP = 1, LB = 95%, PB = 53%) sister to a strongly supported clade (PP = 1, LB = 100%, PB = 96%) including B. tomentosus and species of B. sects. Bromus and Genea, neither of which is monophyletic. Species of B. sect. Bromus and B. sect. Genea form a moderately to strongly supported lineage (PP = 1, LB = 96%, PB = 74%) comprising a five-lineage polytomy: (i) a strongly supported subclade (PP = 1, LB = 100%, PB = 100%) of B. rubens and B. madritensis; (ii) a strongly supported subclade (PP = 1, LB = 100%, PB = 96%) of B. rigidus and B. diandrus; (iii) B. alopecuros subsp. caroli-henrici; (iv) a strongly supported subclade (PP = 0.99, LB = 99%, PB = 94%) in which B. sterilis is sister to a moderately to strongly supported lineage (PP = 99, LB = 94%, PB = 83%) comprising a three-lineage polytomy: B. tectorum subsp. tectorum, B. rechingeri, and a weakly to moderately supported clade (PP = 0.67, LB = 85%) of B. sewerzowii, B. pulchellus, B. oxyodon, and B. gedrosianus; (v) a strongly supported subclade (PP = 0.98, LB = 93%, PB = 74%) of most B. section Bromus species. Within this last clade, all B. scoparius samples form a weakly supported clade (PP = 0.74, PB = 61%) sister to a weakly supported clade (PP = 0.54, LB = 68%) comprising the remaining members of B. sect. Bromus, which form a polytomy.

Although relationships among most B. sect. Bromus species are unresolved, some subclades are present. These subclades include (i) B. squarrosus; (ii) B. pseudobrachystachys; (iii) a strongly supported subclade (PP = 0.96, LB = 100%, PB = 86%) comprising two B. japonicus samples; (iv) a weakly to strongly supported subclade (PP = 0.89, LB = 97, PB = 63%) comprising one B. japonicus sample and one B. intermedius sample; (v) a weakly to strongly supported subclade (PP = 0.81, LB = 94%) comprising B. squarrosus and B. hordeaceus; (vi) a moderately supported subclade (PP = 0.84, LB = 99%, PB = 65%) comprising B. racemosus and B. arvensis; (vii) a weakly supported subclade comprising three B. briziformis samples (PP = 0.67, LB = 97%, PB = 54%); (viii) a moderately to strongly supported subclade (PP = 0.84, LB = 98%, PB = 81%) comprising one B. interruptus sample sister to a clade (PP = 0.89, LB = 98%, PB = 60%) of most B. hordeaceus samples; (ix) a weakly to strongly supported subclade (PP = 0.61, LB = 96%, PB = 61%) comprising samples of B. secalinus, B. racemosus, and B. commutatus sister to a moderately supported subclade (PP = 0.87, LB = 98%, PB = 64%) comprising four samples of B. commutatus and B. racemosus.

Discussion

Our phylogenetic analyses, based on four molecular markers, broad taxon sampling across the genus, and multiple individuals of many species, confirm multiple aspects of Bromus phylogeny identified previously and provide new insights into relationships among the major Bromus lineages, among species within B. sect. Bromus, and among species of B. sects. Bromus and Genea. Our nrDNA and plastid trees support the monophyly of Bromus s.l., consistent with other phylogenetic studies that sampled the genus broadly (Pillay & Hilu, 1995; Saarela et al., 2007; Pourmoshir, Amirahmadi & Naderi, 2019). However, the plastid and nrDNA data resolve relationships among most major lineages of Bromus and among multiple species within major lineages differently. These conflicting topologies usually receive strong support by each of the phylogenetic methods used, consistent with previous studies (Saarela et al., 2007; Fortune et al., 2008; Pourmoshir, Amirahmadi & Naderi, 2019). In most instances the incongruence between nrDNA and plastid trees may be best explained by plastid capture. This occurs when the plastome of one species is introgressed into another species via hybridization followed by backcrossing to the paternal parent, but retaining the maternal parent’s plastome (Rieseberg & Soltis, 1991).

Phylogenetic relationships among major Bromus lineages

Bromus sect. Boissiera, Bromus sect. Nevskiella, and Bromus sect. Mexibromus

Bromus pumilio and B. gracillimus are strongly supported as sister taxa in the nrDNA and plastid trees. These results are consistent with studies based on ITS and ETS data (Alonso, 2015; Pourmoshir, Amirahmadi & Naderi, 2019), and our study is the first one to include plastid data from both species. Long branches subtend both species in the nrDNA tree. These long branches may reflect an accelerated rate of nrDNA evolution in these lineages related to their annual habit (Yue et al., 2010), a long period since the species diversified from their common ancestor, or both. Researchers have not identified macromorphological characters for this two taxon clade.

Bromus pumilio, a diploid (2n = 14; Avdulov, 1931; Smith, 1969), ranges from Egypt to Central Asia, Pakistan, and the Arabian Peninsula (POWO, 2021). It is characterized by an annual habit, spikelets terete, rhachilla internodes about ½ the lemma length, lower lemmas five-awned, upper lemmas five- to nine-awned, and awns flattened, recurved when mature (Smith, 1970). Some researchers have classified B. pumilio in the monotypic genus Boissiera (Boissiera pumilio (Trin.) Stapp [syn. Boissiera squarrosa (Banks & Sol.) Nevski]; e.g., Tzvelev, 1976). Within Bromus, researchers have classified the species in the monotypic B. sect. Boissiera (Smith, 1985b; Naderi & Rahiminejad, 2015) and in B. sect. Bromus (e.g., Smith, 1969, 1970, 1972). Consistent with other studies of DNA sequence data (Grass Phylogeny Working Group II, 2012; Pourmoshir, Amirahmadi & Naderi, 2019), our results support inclusion of B. pumilio in Bromus rather than Boissiera. Indeed, if we were to recognize the species in the genus Boissiera, Bromus would be paraphyletic. Studies based on data from chromosomes and seed protein serology (Smith, 1969, 1972) and allozymes (Oja & Jaaska, 1998) also support classifying B. pumilio in Bromus. Smith (1972) and Stebbins (1981) hypothesized that B. pumilio evolved from within B. subg. Bromus, and morphological studies have suggested a close relationship between B. pumilio and B. danthoniae (B. sect. Bromus), based on the shared character state of multiple lemma awns, which is absent in other Bromus species (Naderi et al., 2016; Pourmoshir, Amirahmadi & Naderi, 2017). Our results do not support either of these hypotheses. Because B. sect. Bromus (including B. danthoniae) and B. pumilio arose independently, the presence of multiple lemma awns in B. pumilio and B. danthoniae is homoplasy.

Bromus gracillimus, a diploid (2n = 14), ranges from eastern Turkey to Xinjiang and the Western Himalayas (POWO, 2021). It is characterized by an annual habit; spikelets few-flowered, up to 10 mm long, terete when young, later somewhat compressed, ovate-lanceolate, and cuneiform when mature; glumes narrow, the lower one-nerved, the upper three-nerved; lemmas rounded on the back; awns single, slender, straight, and four to six times the lemma length (Smith, 1970). Researchers have classified B. gracillimus in B. sect. Nevskiella (Smith, 1985b; Naderi & Rahiminejad, 2015), B. subg. Nevskiella (Stebbins, 1981; Acedo & Llamas, 1999; Llamas & Acedo, 2019), and the monotypic genus Nevskiella (N. gracillima (Bunge) V.I.Krecz. & Vved.; e.g., Tzvelev, 1976). Our nrDNA and plastid results support classifying B. gracillimus in Bromus, consistent with earlier studies (Alonso, 2015; Pourmoshir, Amirahmadi & Naderi, 2019). Classifying the taxon in the genus Nevskiella would result in Bromus being paraphyletic.

In addition to the macromorphological characters that distinguish Bromus pumilio and B. gracillimus from each other and from other Bromus species, researchers have identified unique variation in micromorphological characters within these taxa. In a study of lemma and palea micromorphological variation among 77 Bromus species representing six subgenera/sections, Acedo & Llamas (2001) found the micromorphological characters of B. gracillimus to be most different than other species. They did not study B. pumilio. Similarly, Mosaferi & Keshavarzi (2021) studied lemma and palea micromorphological variation among Bromus species in Iran. In their phenetic analyses of micromorphological data, B. gracillimus formed a cluster separate from all other Bromus sections and B. pumilio formed the first subcluster in a cluster including the rest of the genus except B. gracillimus. Cladistic study is required to determine if any of the lemma and palea micromorphological characters are synapomorphies for the B. gracillimus–B. pumilio clade.

Given the absence of putative morphological synapomorphies for the B. pumilio–B. gracillimus lineage, we believe classification of each species in a subdivisional taxon at the same rank as the other major generic lineages is more appropriate than combining them into a single taxon. The latter approach, however, would be equally consistent with the phylogeny. At sectional rank, B. sect. Boissiera is the correct name for the lineage comprising B. pumilio. At the subgeneric rank, no validly published name for this lineage exists. Stebbins (1981) used the designation “Bromus subg. Boissiera,” but he did not validly publish it. Sectional and subgenus names are available for the lineage comprising B. gracillimus. If we classify the two species in a single higher-level taxon, the names B. subg. Nevskiella and B. sect. Nevskiella have priority at those ranks.

The relationship of the B. pumilio–B. gracillimus clade to other Bromus lineages differs between the plastid and nrDNA trees. In the nrDNA trees, B. pumilio–B. gracillimus and B. densus are sister groups. Bromus densus is one of three species endemic to Mexico that Saarela, Peterson & Valdés-Reyna (2014) classified in B. sect. Mexibromus. They recognized this new sectional taxon based on molecular analyses that identified a strongly supported clade comprising three species previously classified in B. sect. Bromopsis (B. attenuatus Swallen, B. densus, and B. dolichocarpus Wagnon; Wagnon, 1952) that was sister to the rest of the genus (Saarela et al., 2007). The sister-group relationship between B. sect. Mexibromus and B. pumilio–B. gracillimus in our nrDNA trees is consistent with Pourmoshir, Amirahmadi & Naderi’s (2019) nrDNA tree. However, support for this topology in our Bayesian analyses (PP = 0.83), based on combined ITS + ETS data, is weaker than in theirs, based only on ITS data (PP = 1.00). Lower support for the clade here may be due to differences in the ITS alignments, contributions of the ETS data to our phylogenetic results, our inclusion of one species to represent B. sect. Mexibromus (they sampled two species, splitting the long branch subtending B. densus in the nrDNA tree), or a combination of these. Researchers have not previously hypothesized a close relationship among B. pumilio, B. gracillimus, and species now included in B. sect. Mexibromus, nor are we aware of putative morphological synapomorphies for the clade. Study of lemma and palea micromorphological characters in B. sect. Mexibromus species may be insightful, since these characters are distinct in B. pumilio and B. gracillimus compared to the rest of the genus (Acedo & Llamas, 2001; Mosaferi & Keshavarzi, 2021).

In the plastid trees, the B. pumilio–B. gracillimus lineage and B. sect. Ceratochloa are sister groups, consistent with a B. sect. Ceratochloa–B. sect. Neobromus–B. pumilio clade in Pourmoshir, Amirahmadi & Naderi’s (2019) plastid tree. The B. pumilio–B. gracillimus–B. sect. Ceratochloa clade and B. densus are successive sisters to the rest of the genus in our plastid tree. However, in the plastid trees in Pourmoshir, Amirahmadi & Naderi (2019), relationships are unresolved among B. densus, B. dolichocarpus, a B. pumilio–B. sect. Ceratochloa clade, and a clade comprising the rest of the genus. If B. sect. Mexibromus is more closely related to B. pumilio–B. gracillimus than to the rest of the genus, as in the nrDNA trees, ancient hybridization and plastid capture could explain its position in the plastid tree. In this scenario, the B. sect. Mexibromus plastome may have originated in an ancestor of the clade comprising the rest of the genus and introgressed into the B. sect. Mexibromus lineage.

Bromus sect. Ceratochloa and Bromus sect. Neobromus

In the plastid trees, B. sect. Ceratochloa is monophyletic and distantly related to B. sects. Bromus, Bromopsis, and Genea, consistent with other studies (Pillay & Hilu, 1990, 1995; Saarela et al., 2007; Pourmoshir, Amirahmadi & Naderi, 2019). In addition, B. sect. Ceratochloa and B. sect. Neobromus, for which we obtained only nrDNA, are sister taxa in plastid and nrDNA trees (Pillay & Hilu, 1990, 1995; Saarela et al., 2007; Pourmoshir, Amirahmadi & Naderi, 2019). In the nrDNA trees, by contrast, the B. sect. Ceratochloa–B. sect. Neobromus lineage is nested within a large clade including Eurasian B. sect. Bromopsis species except B. ramosus, B. sect. Genea, and B. rechingeri (B. sect. Bromus). This topology is consistent with ITS trees in Saarela et al. (2007) and Pourmoshir, Amirahmadi & Naderi (2019).

Polyploidy challenges reconstructing the history of B. sect. Ceratochloa species. Bromus sect. Ceratochloa comprises hexaploids, octoploids, and dodecaploids; diploids or tetraploids are unknown (Stebbins, 1981; Klos et al., 2009). The taxon includes at least four extant lineages. Two of these lineages are species complexes differing in ploidy, morphology, and geographical distribution: the B. catharticus complex, which comprises hexaploids native to South America, and the B. carinatus complex, which comprises octoploids native to North America. The hexaploids contain three genomes represented by uniform medium-sized chromosomes that Stebbins & Tobgy (1944) designated A, B, and C; the species have AABBCC genomes. The octoploids contain the same three uniform medium-sized genomes (that is, 42 medium-sized chromosomes; AABBCC) and one larger genome (14 large chromosomes) designated LL, putatively derived independently from one or more species of B. sect. Bromopsis via intersectional hybridization (Stebbins & Tobgy, 1944; Stebbins, 1981); the octoploid species have AABBCCLL genomes. Bromus arizonicus (Shear) Stebbins, a duodecaploid native to North America with 96 medium-sized chromosomes (Klos et al., 2009), represents a third lineage. It may be an allopolyploid derived from B. catharticus and B. berteroanus (2n = 6x; B. sect. Neobromus) or a relative of B. berteroanus (Stebbins, Tobgy & Harlan, 1944; Klos et al., 2009). Klos et al. (2009) identified two duodecaploid plants from South America with medium-sized and long chromosomes (thus differing considerably from North American duodecaploid B. arizonicus). They hypothesized that these plants, which are morphologically similar to B. ayacuchensis Saarela & P.M.Peterson (Saarela, Peterson & Refulio-Rodríguez, 2006), have a unique origin; they may represent a fourth lineage of B. sect. Ceratochloa.

Based on the affinities of B. sect. Ceratochloa in the plastid trees, we infer that the plastomes of B. sect. Ceratochloa (and of B. sect. Neobromus, its sister taxon) originated in a common ancestor of B. pumilio and B. gracillimus and became incorporated into ancestral members of the Ceratochloa lineage via introgressive hybridization. The monophyly of B. sect. Ceratochloa in the plastid trees indicates the plastomes of species in the section share a common origin. Accordingly, if the octoploid and duodecaploid members of the section arose via hybridization between members of the hexaploid B. catharticus complex and other Bromus lineages, as researchers have hypothesized, we infer, based on the plastid topology, that individuals of the B. catharticus complex were the maternal parents (plastome donors) in the crossing events. Indeed, if the plastome donor(s) were from Bromus lineages other than B. sect. Ceratochloa, we would not expect B. sect. Ceratochloa to be monophyletic in plastid-based phylogenetic analyses. Given the nrDNA results, we infer that species of B. sect. Bromopsis, their relatives, or their ancestors contributed at least one genome to the B. sect. Ceratochloa–B. sect. Neobromus lineage (Saarela et al., 2007). We do not know, however, if the current nrDNA sequence data represents the A, B, C, or L genomes of B. sect. Ceratochloa species. Resolving this question will require approaches that can identify and reconstruct the histories of the multiple genomes present in B. sect. Ceratochloa, B. sect. Neobromus, B. sect. Bromopsis species from Eurasia, and possibly other Bromus taxa (e.g., Diaz-Perez et al., 2018; Hasterok, Wang & Jenkins, 2020).

Bromus sect. Bromopsis

Bromus section Bromopsis comprises approximately 60 species with various ploidy levels (Stebbins, 1981). Although a comprehensive analysis of B. sect. Bromopsis was beyond the scope of our study, we sampled several species, including some not previously included in molecular phylogenetic studies: B. induratus (2n = 42; Ghukasyan, 2004), B. kopetdagensis (2n = 42, 70; Sheidai et al., 2008), B. pannonicus Kumm. & Sendtn. (2n = 28; Pustahija et al., 2013), B. sclerophyllus (2n = 14, 42, 56; Kozuharov, Petrova & Ehrendorfer, 1981; Strid, Franzen & Love, 1981; Strid, 1983), B. tomentellus (2n = 14, 42, 70, 84) (Mirzaie-Nodoushan et al., 2006; Petrova, Kožuharov & Ehrendorfer, 1997; Sheidai et al., 2008), and B. tomentosus (2n = 28, 42, 84, 156; Mirzaie-Nodoushan et al., 2005; Sheidai et al., 2008). Most of these taxa occur in Southwest Asia (Naderi & Rahiminejad, 2015; POWO, 2021). Bromus sect. Bromopsis is not monophyletic in our analyses, consistent with previous molecular studies (Saarela et al., 2007; Pourmoshir, Amirahmadi & Naderi, 2019). Our nrDNA analyses identify a large clade comprising species of B. sect. Bromopsis, a B. sect. Ceratochloa–B. sect. Neobromus lineage, and a B. sect. Genea–B. rechingeri lineage. Within this large clade, our Bayesian and maximum likelihood analyses identify a strongly supported clade comprising four B. sect. Bromopsis species (B. tomentellus, B. sclerophyllus, B. induratus, B. kopetdagensis) and the B. sect. Genea–B. rechingeri lineage. We find poor support for relationships among B. induratus, B. sclerophyllus, and B. kopetdagensis, whereas B. tomentellus, a species native to Southwest Asia and the eastern Mediterranean countries, is sister to the B. sect. Genea–B. rechingeri lineage. The close relationship among the Old World species B. ramosus and the North American species B. vulgaris and B. kalmii is consistent with morphological and cytological similarities among B. ramosus and North American B. sect. Bromopsis species identified by Armstrong (1983).

In contrast to the nrDNA trees, all sampled New and Old World B. sect. Bromopsis species except B. tomentosus form a clade in the plastid trees, consistent with previous plastid results (Saarela et al., 2007). Bromus tomentosus ranges from eastern Turkey to Pakistan (POWO, 2021). The two B. tomentosus individuals we sampled resolve as sister to the clade comprising species of B. sect. Bromus and B. sect. Genea. Given the discordant affinities of B. tomentosus in the plastid and nrDNA trees, we infer the species likely obtained its plastid genome via plastid capture. Within the B. sect. Bromopsis plastid-based clade, we find strong support for a subclade comprising B. tomentellus, B. induratus, B. sclerophyllus, and B. kopetdagensis. These species are also closely related in the nrDNA trees, whereas other relationships within the clade are either unresolved or poorly supported.

Affinities of Bromus sect. Bromus and Bromus sect. Genea

The nrDNA and plastid data are incongruent regarding the affinities of B. sect. Bromus and B. sect. Genea. The distant relationship between these two sections in the nrDNA phylogeny suggests they have distinct origins, consistent with a lack of chromosomal affinity between them (Stebbins, 1981). In the nrDNA trees, the clade corresponding to B. sect. Bromus (including all sampled species of the section except B. rechingeri) is sister to a broad clade including B. sect. Bromopsis (paraphyletic), B. sect. Ceratochloa–B. sect. Neobromus, and a B. sect. Genea–B. rechingeri clade that is nested deep within the broader lineage. This topology is consistent with ITS trees in Saarela et al. (2007) and Pourmoshir, Amirahmadi & Naderi (2019). In the plastid trees, by contrast, all sampled species of B. sect. Bromus and B. sect. Genea form a clade, consistent with plastid trees in previous studies (Saarela et al., 2007; Fortune et al., 2008; Pourmoshir, Amirahmadi & Naderi, 2019), and neither section is monophyletic. Indeed, the B. sect. Bromus–B. sect. Genea clade comprises a five-lineage polytomy: B. diandrus–B. rigidus (B. sect. Genea), B. madritensis–B. rubens (B. sect. Genea), B. alopecuros subsp. caroli-henrici (B. sect. Bromus), a clade comprising five B. sect. Bromus species (B. gedrosianus, B. oxyodon, B. pulchellus, B. rechingeri, B. sewerzowii) and two B. sect. Genea species (B. tectorum, B. sterilis), and a clade comprising the remaining B. sect. Bromus species.

The Bromus pectinatus complex and allies

Three of the five B. sect. Bromus species that form a clade with B. sterilis and B. tectorum in the plastid tree (B. gedrosianus, B. pulchellus, B. rechingeri) but are part of the B. sect. Bromus lineage in the nrDNA trees are members of the B. pectinatus complex, a group defined by Scholz (1981) that is centered in Southwest Asia and ranges from north Africa to Tibet. The complex comprises morphologically similar tetraploid species characterized by lemmas papery in texture, with prominent nerves and rounded to very bluntly angled margins, lemma apices narrow, bifid into acute teeth or subulate tips usually more than 0.5 mm long, rarely entire, and lower leaf sheaths glabrous to sparsely pilose with rather rigid hairs, or loosely villose with scattered long, soft hairs (Scholz, 1981). Scholz (1981) recognized six species in the complex. More recently, Naderi & Rahiminejad (2015) recognized four (see Introduction). The discordant affinities of B. gedrosianus and B. pulchellus between the nrDNA and plastid trees suggest the possibility of plastid capture from the B. sect. Genea lineage into these species. These results are consistent with the hypothesis that B. pectinatus complex species are derived from hybridization between species of B. sect. Bromus and B. sect. Genea (Stebbins, 1956, 1981; Scholz, 1981). By contrast, our data do not support Tzvelev’s (1976) suggestion that B. gedrosianus may originate from a cross between B. oxyodon and B. racemosus. Our results are, however, consistent with Tzvelev’s (1976) observation that B. pulchellus (as B. tytthanthus) occupies a “somewhat intermediate position” between B. sect. Bromus and B. sect. Genea, indicative of a hybrid origin for the taxon. Previous studies identified the same discordant pattern between nrDNA and plastid data for B. pectinatus—the only B. pectinatus complex species previously included in DNA sequence-based phylogenetic analyses—that we found for B. gedrosianus and B. pulchellus (Saarela et al., 2007; Pourmoshir, Amirahmadi & Naderi, 2019). As in the two previous studies, the two B. pectinatus samples we included in the nrDNA analyses are part of the lineage corresponding to B. sect. Bromus, and relationships among B. pectinatus and other species are poorly resolved and supported. Nevertheless, B. gedrosianus, B. pectinatus, and B. pulchellus do not form a clade in the nrDNA trees, which supports Scholz’s (1981) hypothesis that the origins of the B. pectinatus complex may have involved multiple hybridization events.

Bromus sewerzowii and B. oxyodon are the other two species that form a clade with B. sterilis and B. tectorum (B. sect. Genea) in the plastid trees but are part of the B. sect. Bromus lineage in the nrDNA trees. Accordingly, we infer that B. sewerzowii and B. oxyodon likely obtained their plastomes via plastid capture from the B. sect. Genea lineage, like species of the B. pectinatus complex. Bromus sewerzowii is a tetraploid (2n = 28; Scholz, 1998) that occurs in Iran, Central Asia, Afghanistan, and China (Naderi & Rahiminejad, 2015). It is characterized by an annual habit, panicles 10–17 cm, lower glumes 7–7.5 mm, upper glumes up to 10.5 mm, and lemmas 9–12 mm and lanceolate (Naderi & Rahiminejad, 2015). Bromus sewerzowii is morphologically similar to B. hordeaceus and B. scoparius. These three species have dense panicles with branches and pedicels shorter than the spikelets (Naderi et al., 2012). Despite their morphological similarities, however, our results show they are not closely related. Bromus oxyodon is also a tetraploid (2n = 28; Keshavarzi, Direkvandi & Khoshnood, 2016). It occurs in Afghanistan, NW India, Iran, Kashmir, Kazakhstan, Kyrgyzstan, W Mongolia, Pakistan, Tajikistan, and Uzbekistan (Liu, Zhu & Ammann, 2006; Naderi & Rahiminejad, 2015). Bromus oxyodon is characterized by an annual habit, spikelets oblong-lanceolate and ca. 6 mm wide, lemma apices deeply toothed, with teeth (1.5–)3–4 mm, panicles open, with nodding branches several times longer than the spikelets, lower glumes ca. 10 mm, lemmas 15–18 mm, and awns 20–25 mm, lower part slightly flattened, twisted, and recurved (Liu, Zhu & Ammann, 2006). Bromus oxyodon is morphologically similar to B. lanceolatus (Cope, 1982; Naderi et al., 2016; Pourmoshir, Amirahmadi & Naderi, 2017), but our data do not support a close relationship between them. Bromus oxyodon and B. sewerzowii form a clade with B. gedrosianus and B. pulchellus, both part of the B. pectinatus complex, in the plastid trees, indicating that the plastomes of these four species have a common origin. These results support Scholz’s (1981) idea, based on morphological characters, that B. sewerzowii and B. oxyodon are related to the B. pectinatus complex. The sister group relationship between B. oxyodon and B. pulchellus and the weakly supported clade including B. sewerzowii, B. danthoniae, B. briziformis, and B. gedrosianus (B. pectinatus complex) in the nrDNA trees provides further support for Scholz’s (1981) hypothesis.

The affinities of B. rechingeri in the plastid trees are similar to other B. pectinatus complex species, but in the nrDNA trees B. rechingeri falls in the B. sect. Genea clade rather than the B. sect. Bromus clade like the other B. pectinatus complex species. Given this unexpected nrDNA topology, two of us (A. Nasiri and B. Hamzeh’ee) reviewed the B. rechingeri voucher specimen, housed in TARI, to confirm its identification. The specimen bears multiple annotations. It was originally determined as B. japonicus (without identifier name). Later, M. Alemi identified it as “B. gedrosianus (syn. B. rechingeri)” and R. Naderi as Bromus cf. rechingeri. Using the key in the Bromus treatment in Flora Iranica (Bor, 1970), we identified the specimen as B. rechingeri. The placement of this sample within the B. sect. Genea lineage in the nrDNA trees may indicate a hybrid origin for the specimen (this could explain why Naderi determined the specimen as B. cf. rechingeri) and, perhaps, the species. In addition, this B. rechingeri sample is distinct from B. pulchellus in the nrDNA and plastid trees; the species are not resolved as sister taxa. These results do not support Naderi & Rahiminejad’s (2015) treatment of the name B. rechingeri as a synonym of B. pulchellus. Given that we sampled only one individual of the taxon, future work should sample multiple individuals of B. rechingeri sensu Scholz (1981) to assess and confirm the current results. Future work should also assess the status of B. rechingeri var. afghanicus H.Scholz (=B. rechingeri subsp. afghanicus (H.Scholz) H.Scholz; Scholz, 2008b), which differs from the nominate variety by lemmas 11–17 mm, awns up to 22 mm and inserted 6–10 mm below the lemma apices, and lemma teeth up to 4 mm. Scholz (1981) suggested this taxon may be transitional between B. rechingeri subsp. rechingeri and B. oxyodon based on its morphology, whereas Naderi & Rahiminejad (2015) placed it in synonymy of B. pulchellus.

Researchers have also suggested B. arenarius, a tetraploid native to Australia, may be an intersectional amphidiploid originating from hybridization between B. sect. Bromus and B. sect. Genea species based on chromosomal data derived from crossing experiments and morphology (Knowles, 1944; Stebbins, 1956, 1981). Stebbins (1981) suggested B. arenarius is morphologically intermediate between these sections, based on the number of veins in the glumes and lemmas, the shape of its lemmas, which taper apically more than is typical in B. sect. Bromus, and the shape of its palea apex (truncate in B. sect. Bromus species, strongly bidentate in B. sect. Genea species, and weakly bidentate in B. arenarius). Additionally, B. arenarius and B. pectinatus are morphologically similar, and some authors have suggested the two may not be distinct species (e.g., Clayton, 1971). We included in our analyses previously published ITS and ETS data for B. arenarius, and in the nrDNA trees the taxon is part of a subclade including B. pectinatus within Bromus sect. Bromus clade C. These results are consistent with Ainouche & Bayer (1997), who found a close relationship between B. arenarius and B. adoensis Hochst. ex Steud. (=B. pectinatus) in their ITS trees. Based on their ITS phylogeny, Ainouche & Bayer (1997) rejected the hypothesis of B. arenarius being a hybrid between B. sects. Bromus and Genea. Because data for the two plastid regions sampled in our study was not available for B. arenarius, we were unable to test the hybrid-origin hypothesis for the taxon by comparing nrDNA and plastid topologies that include the taxon. Nevertheless, other plastid data for B. arenarius provide some insight into its likely affinities in plastid-based phylogenies. A rbcL sequence from a B. arenarius accession collected in New Zealand (AY691632.1, R.C. Garder et al., unpublished data, 2016) is more similar to rbcL sequences of B. sterilis (B. sect. Genea) than of B. sect. Bromus species, based on a GenBank BLAST search. The rbcL similarity between B. arenarius and B. sect. Genea is consistent with the topology we observed for B. gedrosianus, B. oxyodon, B. pulchellus, B. rechingeri, and B. sewerzowii in our plastid trees. We thus infer that nrDNA and plastid regions in B. arenarius are incongruent, supporting the possibility of an intersectional hybrid origin for the taxon, as first hypothesized over 75 years ago.

Our results provide insight into the possible identities of B. sect. Genea taxa from which the plastomes of the B. pectinatus complex and allies likely originated. Based on the plastid topology, we infer that B. sterilis (2n = 14, 28; e.g., Sheidai & Fadaei, 2005), B. tectorum (2n = 14; e.g., Sheidai & Fadaei, 2005), or ancestors of one or both species were plastome donors to the lineage including B. gedrosianus, B. oxyodon, B. pulchellus, and B. sewerzowii. These results are consistent with Sales (1993) suggestion, based on morphological data, that B. tectorum is the “link” between the B. pectinatus complex and B. sect. Genea. Although our results provide new insights into the putative origins of the intersectional amphidiploid Bromus species, we need additional research based on unlinked nuclear markers to determine more precisely the taxa of B. sect. Bromus and B. sect. Genea putatively involved in their origins. Future work should also sample B. tibetanus, a member of the B. pectinatus complex that researchers have omitted from molecular studies. Additionally, we need further study of B. pseudojaponicus sensu Scholz (1981), to determine whether the name corresponds to a distinct lineage, given that Naderi & Rahiminejad (2015) treated the name as a synonym of B. pulchellus.

Phylogenetic relationships within B. sect. Bromus

We identify four main lineages within B. sect. Bromus that comprise one, two, or multiple species: B. alopecuros subsp. caroli-henrici, B. hordeaceus and B. interruptus, B. scoparius, and a clade (clade C) comprising all other sampled species of the section. The long branch in the nrDNA tree subtending B. sect. Bromus clade A, which includes all taxa except B. alopecuros subsp. caroli-henrici, indicates considerable molecular evolution in the nrDNA regions since the lineage split from its sister group, B. alopecuros subsp. caroli-henrici, and before the lineage diversified. Similarly, a long branch in the nrDNA tree separates the B. hordeaceus–B. interruptus clade from B. sect. Bromus clade B, indicating considerable genetic differentiation between these lineages, and the B. scoparius clade is subtended by a slightly longer branch compared to its sister lineage, B. sect. Bromus clade C. Within clade C, however, branch lengths are short in the nrDNA tree. This is indicative of little variation in the gene regions we sampled. Indeed, phylogenetic resolution in this clade is poor and only three species (B. gedrosianus, B. oxyodon, B. pulchellus) are resolved as monophyletic. Aside from the strongly supported three-taxon clade comprising B. briziformis, B. danthoniae, and B. sewerzowii, our nrDNA analyses identify a few clades comprising individuals of multiple species, three of which are strongly supported in Bayesian and maximum likelihood analyses, whereas relationships among most individuals and species in the clade are unresolved (i.e., clade C comprises a large polytomy). We are unable to make inferences about phylogenetic relationships among most species in the clade due to the lack of resolution. The pattern is similar in the plastid trees, where only B. briziformis and B. pseudobrachystachys are resolved as monophyletic, one clade comprises multiple individuals of several species, and relationships among most individuals and multi-individual lineages are unresolved. Low genetic differentiation in the nrDNA and plastid sequences among most B. sect. Bromus species indicates they are likely relatively young and probably diversified rapidly. Moreover, multiple speciation events may have happened at the same time, and some lineages may be equally closely related to one another, challenging phylogenetic reconstruction. We are unable to distinguish between the hypotheses of rapid radiation or multiple simultaneous speciation events with the current data. Nevertheless, our results are consistent with Stebbins’ (1981) idea that many B. sect. Bromus species likely evolved in response to agricultural conditions, including grazing livestock. Similarly, Scholz (2008b) suggested the species may be anecophytes, plant taxa that originated under the influence of human activities that may comprise cultigenic taxa and weeds, without occurrences in natural vegetation.

Aside from lack of sequence variation in the nrDNA regions, the lack of species monophyly in clade C in the nrDNA trees may, in some instances, be due to the presence of multiple rDNA arrays within individuals, incomplete concerted evolution, incomplete lineage sorting, or a combination of these (e.g., Feliner & Rossello, 2007). Indeed, many species that are part of this large clade are polyploids that likely arose from hybridization between closely related B. sect. Bromus species, combining independent genomes, and there were instances of polymorphism in a subset of our nrDNA sequences, which we coded with nucleotide ambiguity codes. Misidentification may also be a factor in some instances, even though we were careful in attempting to ensure all sampled individuals were accurately determined. A limitation of our study is that we did not clone the nrDNA regions we studied to obtain all sequence variants that may be present within closely related individuals and species, many of which are polyploids. Studies of grasses that characterized nrDNA variation within and among individuals of multiple closely related diploid and polyploid species found numerous haplotypes within individuals and species and complicated patterns of nrDNA evolution (e.g., Peng et al., 2010; Fan et al., 2014). The same complicated pattern is likely present within B. sect. Bromus. Given our results, future study of nrDNA in B. sect. Bromus (and other Bromus sections) would benefit from approaches that captures all ribosomal DNA copies within individuals, whether through cloning or next generation sequencing approaches (e.g., Matyášek et al., 2012; Song et al., 2012). Additionally, phylogenomic approaches that sample many loci from across the nuclear genome (e.g., genotyping-by-sequencing) would likely be informative in reconstructing the clade’s evolutionary history, including identifying the parental taxa of the polyploid species. Indeed, such approaches have been successful in resolving relationships among closely related taxa in diverse groups of plants, including grasses (e.g., Guo, Guo & Li, 2019; Hyun et al., 2020). Nevertheless, our results show that direct sequencing of nrDNA is appropriate for resolving some deeper relationships with Bromus s.l. and within B. sect. Bromus.

Affinities of B. alopecuros subsp. caroli-henrici

Bromus alopecuros subsp. caroli-henrici is sister to the clade comprising the rest of B. sect. Bromus (clade A) in the nrDNA trees, consistent with the ITS tree in Ainouche et al. (1999). Bromus alopecuros subsp. caroli-henrici is similarly distinct from other B. sect. Bromus species in the plastid tree. However, additional plastid genome sampling will be necessary to resolve the polytomy in the plastid tree formed by B. alopecuros subsp. caroli-henrici, the clade comprising most B. sect. Bromus species, and the lineages comprising species of B. sect. Genea and B. pectinatus and allies.

These results provide insight into the taxonomy of B. alopecuros Poiret (1789) and B. alopecuros subsp. caroli-henrici. Bromus alopecuros is native to the central and eastern Mediterranean (POWO, 2021), diploid (2n = 14; Ainouche et al., 1999), and morphologically similar to B. lanceolatus (Acedo & Llamas, 1994). Bromus caroli-henrici Greuter (1971) differs from B. alopecuros by panicles 1–2 cm broad (vs panicles 1–3 cm broad), spikelets often borne singly at nodes (vs spikelets mostly 2–4 at each node), and lemmas with acuminate apical teeth (vs lemmas with triangular apical teeth) (Smith, 1985a). Like B. alopecuros, B. caroli-henrici is diploid (2n = 14; Ainouche et al., 1999; Vogt & Aparicio, 1999). It ranges from Greece to southern Turkey and Syria (POWO, 2021). Smith in Heywood (1978) suggested B. caroli-henrici may have originated from isolation of a B. alopecuros s.str. population on an Aegean Island, and he proposed treating it as a subspecies of B. alopecuros. Other authors, however, recognize the taxa at species level (i.e., B. alopecuros and B. caroli-henrici; Pavlick et al., 2003; Saarela & Peterson, 2019). Some authors recognize a third infraspecific taxon, B. alopecuros subsp. biaristulatus (Maire) Acedo & Llamas (e.g., Scholz in Greuter & Raus, 2008; Scholz, 2008b; Valdés et al., 2009), a name Acedo & Llamas (1994) considered synonymous with B. alopecuros subsp. caroli-henrici. Our nrDNA trees show that B. alopecuros s.str. and B. alopecuros subsp. caroli-henrici are not closely related. Bromus alopecuros subsp. caroli-henrici is the sister group of the clade comprising the rest of B. sect. Bromus, and B. alopecuros is part of Bromus sect. Bromus clade C. This topology is consistent with ITS trees in Ainouche et al. (1999) and Pourmoshir, Amirahmadi & Naderi (2019), although the latter study did not sample B. caroli-henrici; each of these studies included independent B. alopecuros s.str. samples. Accordingly, the molecular results support recognition of B. alopecuros and B. alopecuros subsp. caroli-henrici as distinct species. Plants corresponding to B. alopecuros subsp. biaristulatus sensu Scholz should be sampled in future analyses to assess the status of the name.

Affinities of Bromus hordeaceus and allies

The next successive sister to the rest of the B. sect. Bromus clade after B. alopecuros subsp. caroli-henrici is a lineage comprising B. hordeaceus and B. interruptus. In the nrDNA tree, these two species form a strongly supported clade, but neither is monophyletic. In the plastid tree, the sample of B. interruptus from which we obtained data from both plastid regions (accession 1) is sister to a clade comprising all but one B. hordeaceus sample (accession 6). The rpl32-trnL sequences from B. interruptus (accession 1) and the B. hordeaceus samples are identical, whereas the matK sequences in B. interruptus (accession 1) and the B. hordeaceus samples differ by one base. The other B. interruptus sample in the plastid tree, however, is not included in a clade with B. hordeaceus. This sample is represented by a matK sequence (Dataset S2) identical to the B. hordeaceus and B. interruptus matK sequences we generated. Data are missing from the 5′ end of the gene where we detected the single base pair substitution in our B. interruptus accession. This explains the exclusion of the sample from the B. hordeaceus–B. interruptus clade. Similarly, the B. hordeaceus sample that is placed separately from the other samples of the species is missing data from the 5′ end of matK, the only plastid region we included for the sample, where there is a substitution that unites the other samples of the species.

Bromus hordeaceus is a broadly distributed, morphologically variable tetraploid (2n = 28) with multiple infraspecific taxa; Scholz (2008b), for example, recognized seven subspecies. Bromus interruptus, a winter annual grass, is endemic to southern and eastern England and tetraploid (2n = 28; e.g., Maude, 1940; Dempsey, Gornall & Bailey, 1994). It differs from B. hordeaceus by a compacted, interrupted inflorescence, short rachilla internodes, and paleas bifid to the base (Lyte & Cope, 1999; Rich & Lockton, 2002). Bromus interruptus is assessed as Extinct in the Wild (EW) on the International Union for Conservation of Nature (IUCN) Red List (Bilz, 2011), but it persists in cultivation from seed P.M. Smith collected before it disappeared in the wild. M.J. Harvey grew the specimen we sequenced (CAN 605189) in a greenhouse at Dalhousie University (St. John’s, Newfoundland and Labrador, Canada) from seed provided by Royal Botanical Gardens, Kew. The close relationship between B. hordeaceus and B. interruptus in our nrDNA and plastid trees is consistent with seed protein (Smith, 1972), allozyme (Oja, 1998), and other nrDNA data (Ainouche & Bayer, 1997). Data from both genomes supports the hypothesis that B. interruptus is derived from B. hordeaceus, likely via mutation (Smith, 1972; Ainouche & Bayer, 1997).

Molecular sampling of B. incisus and B. lepidus, species many authors consider to be closely related to B. hordeaceus (e.g., Smith, 1970, 1972; Scholz, 2008a; Table 1), is needed to confirm their affinities. Based on protein serology data, Smith (1972) suggested B. lepidus may have evolved from B. brachystachys, i.e., B. brachystachys may be a parent of the tetraploid B. lepidus. Scholz (2008a), however, considered this hypothesis unlikely. In addition, Scholz (2008b) suspected B. incisus, a tetraploid (2n = 28), to be a hybrid or an introgressive hybridization product of B. lepidus and B. hordeaceus s.l. that arose no more than about 200 years ago. Some authors treat the name B. incisus as a synonym of B. lepidus (WCVP, 2021; Englmaier & Wilhalm, 2018). Bromus parvispiculatus H.Scholz, which Scholz (2008b) described from southern Europe as a member of the B. hordeaceus alliance, is likely part of the B. hordeaceus–B. interruptus clade; this also requires confirmation with molecular data. Since its description, authors have treated this name as a synonym of B. hordeaceus subsp. hordeaceus (WCVP, 2021) or have recognized the taxon as B. hordeaceus subsp. parvispiculatus (H.Scholz) Barina (Barina et al., 2018).

In the nrDNA trees, the sister-group relationship between B. hordeaceus–B. interruptus and Bromus sect. Bromus clade B, which comprises the remainder of B. sect. Bromus except B. alopecuros subsp. caroli-henrici, is consistent with nrDNA trees in Ainouche & Bayer (1997) and Ainouche et al. (1999). Smith (1972) suggested B. hordeaceus may have originated from the crossing of B. arvensis (diploid) and B. scoparius (2n = 14, 28; e.g., Sokolovskaya & Probatova, 1979; Ghukasyan, 2004). However, the deep split between the B. hordeaceus–B. interruptus lineage and Bromus sect. Bromus clade B, which includes B. arvensis and B. scoparius, in the nrDNA tree does not support Smith’s (1972) hypothesis. Instead, the nrDNA topology suggests independent origins for the B. hordeaceus–B. interruptus lineage and Bromus sect. Bromus clade B. Based on the same nrDNA topology, Ainouche & Bayer’s (1997) rejected Smith (1972) hypothesis and suggested a diploid parent of B. hordeaceus, which they assumed is an allopolyploid, may have been an extinct or undiscovered species. Scholz (2008a) considered this scenario to be unlikely and instead suggested B. hordeaceus may be autopolyploid. Further work is needed to test both these hypotheses. Unlike the nrDNA tree, the plastid tree indicates B. hordeaceus and B. interruptus are closely related to all other B. sect. Bromus species, although the relationships of these two species to others in the clade are unresolved. Based on the nrDNA and plastid topologies, we infer that the B. hordeaceus–B. interruptus lineage may have obtained its plastome via plastid capture from a species of the Bromus sect. Bromus clade B in the nrDNA tree.

Affinities of Bromus danthoniae

Bromus danthoniae occurs in Turkey, Iran, Caucasus, Russia, Central Asia, Afghanistan, Pakistan, NW India, Iraq, Syria, Lebanon, Palestine, the Arabian Peninsula, and China, and it differs from other Bromus species by lemmas with 3–5 awns (rarely 1, 2, or 6) (Naderi & Rahiminejad, 2015). Because of unique morphological characteristics, determining the affinities of B. danthoniae has long challenged taxonomists. Indeed, researchers have classified B. danthoniae in B. sect. Triniusia, in the monotypic genus Triniusa Steud. (T. danthoniae Steud.; Tzvelev, 1976), and in B. sect. Bromus/B. subg. Bromus (Smith, 1970; Stebbins, 1971). In the plastid and nrDNA trees, B. danthoniae is nested in the clade corresponding to B. sect. Bromus, consistent with previous studies (Ainouche & Bayer, 1997; Saarela et al., 2007). Scholz (1998) recognized three subspecies within B. danthoniae (B. danthoniae subsp. danthoniae, B. danthoniae subsp. pseudodanthoniae (Drobow) H.Scholz, B. danthoniae subsp. rogersii C.E.Hubb. ex H.Scholz), and he also described B. turcomanicus H.Scholz, a second species of B. sect. Triniusia he knew only from the type locality in Turkmenistan. In a revision of these taxa, Naderi et al. (2016) recognized two varieties within B. danthoniae: B. danthoniae var. danthoniae (syn. B. danthoniae subsp. rogersii, B. turcomanicus, B. danthoniae subsp. pseudodanthoniae) and B. danthoniae var. pauciaristatus. The latter taxon differs from the nominate variety by most lemmas having one awn and upper lemmas of at least one spikelet with multiple awns, or one awn with one or more minute awns.

We followed the taxonomy of B. danthoniae proposed by Naderi et al. (2016) and sampled multiple individuals of both varieties and some individuals identified only to species level. In the nrDNA trees, B. danthoniae var. danthoniae, B. danthoniae var. pauciaristatus, B. briziformis, and B. sewerzowii form a strongly supported clade. Of these taxa, only B. sewerzowii, for which we sampled two individuals, is recovered as monophyletic. Its affinity with B. briziformis and B. danthoniae based on nrDNA data is a novel result, as the species has not previously been sampled in a molecular phylogenetic analysis. Previous researchers did not consider B. sewerzowii to be closely related to B. briziformis or B. danthoniae (Kreczetovich & Vvedensky, 1934; Pénzes, 1936). Relationships among the individuals of the other three taxa are unresolved. These results differ from Pourmoshir, Amirahmadi & Naderi’s (2019) ITS tree, in which the three B. danthoniae individuals they sampled, representing both varieties, formed a weakly supported clade. The lack of resolution among the B. danthoniae samples in the nrDNA trees here neither supports nor refutes the taxonomy of B. danthoniae proposed by Naderi et al. (2016). In the plastid trees here and in Pourmoshir, Amirahmadi & Naderi (2019), B. danthoniae is also not resolved as monophyletic.

Ainouche & Bayer (1997) and Pourmoshir, Amirahmadi & Naderi (2019) also found a close relationship between B. danthoniae and B. briziformis, both diploids, based on ITS data. Researchers have not, however, hypothesized a close relationship between these species based on morphology, given their dissimilar appearance. Bromus briziformis is a diploid (2n = 14; e.g., Sokolovskaya & Probatova, 1979) native to Southwest Asia and Europe. It differs from B. danthoniae by lemmas inflated and lacking awns or with minute awns up to 3 mm (vs lemmas strongly laterally compressed with 3–5 awns, rarely 1, 2, or 6) and spikelets broadly ovate (vs spikelets ovate or oblongate-lanceolate) (Scholz, 1998; Naderi & Rahiminejad, 2015). Smith (1970) placed B. briziformis in a group with B. japonicus and B. squarrosus based on morphological similarity. Considering the phylogenetic results, B. briziformis and B. danthoniae should be re-examined to search for non-molecular synapomorphies.

Several authors have considered the allotetraploid B. lanceolatus (2n = 28), distributed from the Mediterranean to Xinjiang and Pakistan, and B. danthoniae to be closely related (Kreczetovich & Vvedensky, 1934; Scholz, 1970; Smith, 1972; Naderi et al., 2016; Table 1), based on similarities in their spikelet characteristics. Furthermore, Smith (1972) thought B. danthoniae may be a diploid ancestor of B. lanceolatus, based on protein serology data. Ainouche & Bayer (1997) found B. lanceolatus to be part of an unsupported clade that also included B. briziformis, B. danthoniae, and B. racemosus. Pourmoshir, Amirahmadi & Naderi (2019) found B. lanceolatus to be part of a weakly supported clade with B. alopecuros, B. arvensis, B. brachystachys, B. japonicus, B. intermedius, B. lanceolatus, B. pectinatus, B. secalinus, and B. squarrosus. Our results are consistent with Pourmoshir, Amirahmadi & Naderi (2019). We found B. lanceolatus to be part of a clade with B. pectinatus (2n = 28), B. intermedius (2n = 14; e.g., Ainouche et al., 1999), B. japonicus p.p. (2n = 14; e.g., Ghukasyan, 2004), and B. squarrosus p.p. (2n = 14; e.g., Ghukasyan, 2004). Based on the nrDNA phylogeny, one B. lanceolatus genome donor is likely a diploid species that is part of this clade, but our analyses provide no insight into whether B. danthoniae is a genome donor.

Affinities of other species

Relationships among other species of B. section Bromus are unresolved in the plastid tree, except for the B. hordeaceus–B. interruptus clade and a four-taxon clade comprising B. commutatus samples, all B. secalinus samples except no. 4, for which only matK data was available, the single B. bromoideus accession, and a subset of the B. racemosus samples. The species in the four-taxon clade are tetraploids (Koch et al., 2016). In the nrDNA tree, by contrast, relationships among B. commutatus, B. racemosus, and B. secalinus are unresolved, whereas B. bromoideus is part of a clade also including B. danthoniae, B. briziformis, B. sewerzowii, B. gedrosianus, B. oxyodon, and B. pulchellus; relationships among these taxa are mostly unresolved. The plastid tree indicates the tetraploid taxa, which are allopolyploids, share a common plastid genome donor. If the plastid genomes of B. racemosus and B. commutatus have separate origins, the individuals of B. racemosus in this clade may have obtained their plastomes via plastid capture, given that B. racemosus and B. commutatus hybridize (e.g., Smith, 1973). Alternatively, some plants we sampled as B. racemosus may be misidentified, despite our attempts to ensure that all specimens were identified correctly. Indeed, B. racemosus and B. commutatus can be difficult to distinguish (Smith, 1973; Spalton, 2002).

Bromus bromoideus, B. secalinus, and B. grossus, weedy species associated with cereal crops, are morphologically similar. Bromus bromoideus, endemic to Belgium, became extinct in the wild in the 1930s but persists in cultivation (Koch et al., 2016). Bromus grossus is endemic to Central Europe, and populations are declining across much of its range (Koch et al., 2016). Bromus secalinus is widespread throughout Europe and has been introduced elsewhere (Koch et al., 2016). Smith (1981) hypothesized that B. bromoideus originated from B. secalinus, whereas Koch et al. (2016) suggested, based on AFLP data, that B. bromoideus may have evolved from B. grossus. Koch et al. (2016) also sequenced three plastid regions for these three species and found no sequence variation among them. Our plastid data are consistent with Smith’s (1981) hypothesis of a close relationship between B. bromoideus and B. secalinus. However, unlike in Koch et al. (2016), our sample of B. grossus differs from B. bromoideus and B. secalinus in the plastid regions analyzed and is not part of the clade including these species. Clarifying the affinities of B. grossus based on plastid data will require additional sampling of multiple individuals.

Subdivisional classification and relationships among species of Bromus sect. Bromus

Throughout the last century, researchers have classified species of B. sect. Bromus and B. subg. Bromus in various subdivisional taxa, including sections, subsections, and series, primarily based on patterns of morphological variation (Table 1). None of these classifications have been compared with phylogenetic data to determine if any of their taxa correspond to natural groups. Before interpreting these classifications in the context of our phylogenetic results, we provide an overview of them.

Holmberg (1924) proposed two sections of B. subg. Zeobromus (Griseb.) Hack. (=B. subg. Bromus), circumscribed based on anther length: B. sect. Macrantheri Holmb., with two species, and B. sect. Brachyantheri Holmb., with eleven species. He recognized two subgroups in B. sect. Brachyantheri: B. subsect. Coriacei Holmb., nom. superfl., with five species, and B. subsect. Molles Holmb., with six species. The name B. subsect. Coriacei is superfluous and illegitimate because the subsection as circumscribed by Holmberg includes the type of the genus, B. secalinus.

Researchers proposed three classifications of Bromus s.str. and B. subg. Bromus in the 1930s. Nevski (1934) modified Holmberg’s (1924) classification and recognized three sections within Bromus s.str.: B. sect. Aphaneuroneuron Nevski, nom. superfl., including the seven taxa Holmberg classified in B. sect. Macrantheri and B. subsect. Coriacei; B. sect. Sapheneuron Nevski, including the six taxa Holmberg classified in B. subsect. Molles plus three additional species; and B. sect. Triniusia (Steud.) Nevski, including B. danthoniae. The name B. sect. Aphaneuroneuron is superfluous and illegitimate because the section includes the type of the genus, B. secalinus. In the same year, Kreczetovich & Vvedensky (1934) proposed a classification for B. subg. Zeobromus (=B. subg. Bromus) species in the former Union of Soviet Socialist Republics comprising seven series: B. series Secalini V.I.Krecz. & Vved., nom. superfl., with one species; B. series Macrantherae V.I.Krecz. & Vved., with one species; B. series Squarrosae V.I.Krecz. & Vved., with four species; B. series Commutatae V.I.Krecz. & Vved., with four species; B. series Dolicholepides V.I.Krecz. & Vved., with three species; B. series Ambiguae V.I.Krecz. & Vved., with one species; and B. series Macrostachyae V.I.Krecz. & Vved., with two species. The name B. series Secalini is superfluous because its type, B. secalinus, is the type of the genus. Pénzes (1936) arranged species of B. subg. Serrafalcus (Parl.) Pénzes (=B. subg. Bromus) in four groups based on morphology: “B. sect. Arvenses,” with three species; “B. sect. Racemosi,” with three species; “B. sect. Commutati,” with eleven species; and “B. sect. Pectinati,” with seven species. However, Pénzes (1936) omitted descriptions and diagnoses for his groups thus his designations are not validly published names.

One subdivisional classification of Bromus was proposed in the 1960s and three in the 1970s. Tournay (1961) recognized three subsections of B. sect. Bromus: B. subsect. Coriacei, nom. superfl.; B. subsect. Triniusia (Steud.) Tournay; and B. subsect. Molles. He did not list the species included in each subsection. Scholz (1970) recognized two sections within B. subg. Bromus: B. sect. Triniusia and B. sect. Bromus, and he subdivided B. sect. Bromus into eight series: B. series Intermediae H.Scholz, with one species; B. series Macrantherae, with two species; B. series Squarrosae, with two species; B. series Molles H.Scholz, with three species; B. series Interruptae H.Scholz, with one species; B. series Racemosae H.Scholz, with two species; B. series Secalinae, nom. superfl. (=B. series Bromus), with three species; and B. series Michelariae H.Scholz, with one species. Smith (1972) characterized affinities among groups of species in B. sect. Bromus based on serological evidence of caryopsis proteins and morphological similarity in characters such as lemma shape, lemma texture, panicle shape, and caryopsis size and shape. Using morphological data, he recognized nine groups, designated 1 to 9. Based on the results of his serological experiments, he revised his morphology-based classification slightly and recognized ten groups, designated A to J, each including one to four species. Smith (1972) did not assign scientific names to his species groups. Tzvelev (1976), in his treatment of Soviet Union grasses, recognized three sections within Bromus s.str.: B. sect. Aphaneuroneuron, nom. superfl. (=B. sect. Bromus), with nine species; B. sect. Triniusia, with one species; and B. sect. Sapheneuron, with six species.

Researchers have published classifications of Bromus s.str./Bromus s.l. subg. Bromus in recent decades. Tzvelev (1999), in Flora of Russia: The European Part and Bordering Regions, recognized three sections within Bromus s.str.: (1) B. sect. Bromus, with the same species he (Tzvelev, 1976) had previously included in B. sect. Aphaneuroneuron; (2) B. sect. Triniusia, also equivalent to his earlier treatment (Tzvelev, 1976); and (3) B. sect. Sapheneuron, with one species. Acedo & Llamas (2005) characterized morphological variation in a subset of Iberian Bromus s.l. subg. Bromus species and grouped species into two sections: B. sect. Bromus, with ten species, and a new section, B. sect. Squarrosi Acedo & Llamas, with six species. Scholz (2008a), in a synopsis of Bromus s.str. in Central Europe, Germany, and Austria, recognized three sections: B. sect. Bromus, with eight species; B. sect. Sapheneuron, with three species; and B. sect. Triniusia, with one species (B. danthoniae). Scholz (2008a) treated the names B. sect. Macrantheri, B. subsect. Brachyantheri, and B. sect. Squarrosi as synonyms of the name B. sect. Bromus, and he treated the name B. subsect. Molles as a synonym of B. sect. Sapheneuron.

Although many relationships among B. sect. Bromus species remain unresolved in the nrDNA and plastid trees, there is sufficient resolution in our trees to conclude that few previously recognized subdivisional taxa of B. sect. Bromus/B. subg. Bromus correspond to natural groups. This is particularly true for the nrDNA trees, in which several previously recognized taxa are polyphyletic. These include Bromus subsect. Molles sensu Holmberg (1924), B. sect. Sapheneuron sensu Nevski (1934) and sensu Tzvelev (1976), all the multi-species series recognized by Kreczetovich & Vvedensky (1934) (i.e., B. series Commutatae, B. series Dolicholepides, B. series Squarrosae, and B. series Macrostachyae), B. sect. Aphaneuroneuron and B. sect. Sapheneuron sensu Tzvelev (1976), Bromus groups 7/F, 4/E, and 9/J sensu Smith (1972), B. sect. Bromus sensu Tzvelev (1999), and B. sect. Squarrosi sensu Acedo & Llamas (2005). Bromus sect. Bromus sensu Scholz (1970) is paraphyletic with respect to B. sect. Triniusa. We are unable to assess the monophyly of B. series Macrantherae sensu Scholz (1970) due to unresolved relationships between B. arvensis and B. brachystachys and of B. series Racemosae sensu Scholz (1970) because we did not sample B. pseudosecalinus.

The only previously recognized group within B. sect. Bromus that corresponds to a monophyletic group in our nrDNA trees is Smith’s (1972) Group A, in which he included B. hordeaceus and allies and B. interruptus. By contrast, Scholz (1970) placed B. hordeaceus and allies (taxa he later recognized as subspecies of B. hordeaceus) in B. series Molles and B. interruptus in B. series Interruptae. Our results do not support inclusion of B. interruptus in a group separate from B. hordeaceus. If we were to recognize the lineage including B. hordeaceus as a subsection or a series, the correct names for the taxon at these ranks are likely B. subsect. Molles and B. series Molles; B. hordeaceus is the type of both names. If, in a classification of the genus treating the major generic lineages as subgenera, we were to recognize the clade including B. hordeaceus and B. interruptus as a section, we would require a new name at that rank.

Supplemental Information

Supplemental Information 1 Combined nuclear DNA ITS + ETS dataset.

Click here for additional data file.

Supplemental Information 2 Combined plastid DNA rpl32-trnLUAG + matK dataset.

Click here for additional data file.

Supplemental Information 3 The best substitution models for each subset of sites of nuclear DNA sequences.

Click here for additional data file.

Supplemental Information 4 The best substitution models for each subset of sites of chloroplast DNA sequences.

Click here for additional data file.

Supplemental Information 5 A portion of the majority rule consensus tree inferred from Bayesian analysis of nrDNA ITS sequences.

Bayesian posterior probabilities and maximum likelihood bootstrap support are indicated above the branches, respectively. Posterior probabilities <0.5 and bootstrap support <50% are indicated with a hyphen.

Click here for additional data file.

Supplemental Information 6 A portion of the majority rule consensus tree inferred from Bayesian analysis of nrDNA ETS sequences.

Bayesian posterior probabilities and maximum likelihood bootstrap support are indicated above the branches.

Click here for additional data file.

Supplemental Information 7 A portion of the majority rule consensus tree inferred from Bayesian analysis of cpDNA rpl32-trnLUAG sequences.

Bayesian posterior probabilities and maximum likelihood bootstrap support are indicated above the branches.

Click here for additional data file.

Supplemental Information 8 A portion of the majority rule consensus tree inferred from Bayesian analysis of cpDNA matK sequences.

Bayesian posterior probabilities and maximum likelihood bootstrap support are indicated above the branches.

Click here for additional data file.

Supplemental Information 9 Maximum likelihood tree inferred from nuclear ribosomal ITS + ETS sequence data.

Numbers above branches are bootstrap values.

Click here for additional data file.

Supplemental Information 10 Maximum parsimony tree inferred from nuclear ribosomal ITS + ETS sequence data.

Numbers above branches are bootstrap values.

Click here for additional data file.

Supplemental Information 11 Maximum likelihood tree inferred from plastid DNA rpl32-trnLUAG + matK sequence data.

Numbers above branches are bootstrap values.

Click here for additional data file.

Supplemental Information 12 Maximum parsimony tree inferred from plastid DNA rpl32-trnLUAG + matK sequence data.

Numbers above branches are bootstrap values.

Click here for additional data file.

We thank the curators and staff of several herbaria (TARI, HUI, CAN, W, H) who facilitated access to herbarium specimens and provided leaf materials. We are especially grateful to Dr. Ernst Vitek and Dr. Anton Igersheim, both from W, for their significant support in obtaining and mailing tissue samples. We also thank Dr. Thomas Onuferko for assistance with preliminary analyses. Dr. Carmen Acedo and Dr. Xu Su provided constructive feedback on an earlier version of the manuscript.

Additional Information and Declarations

Competing Interests

Author Contributions

DNA Deposition

Data Availability

The authors declare that they have no competing interests.

Akram Nasiri conceived and designed the experiments, performed the experiments, analyzed the data, prepared figures and/or tables, authored or reviewed drafts of the article, and approved the final draft.

Shahrokh Kazempour-Osaloo conceived and designed the experiments, authored or reviewed drafts of the article, and approved the final draft.

Behnam Hamzehee conceived and designed the experiments, authored or reviewed drafts of the article, and approved the final draft.

Roger D. Bull conceived and designed the experiments, authored or reviewed drafts of the article, and approved the final draft.

Jeffery M. Saarela conceived and designed the experiments, analyzed the data, authored or reviewed drafts of the article, and approved the final draft.

The following information was supplied regarding the deposition of DNA sequences:

Our nucleotide sequences are available at GenBank: OM141017 to OM141113 for nrDNA ITS sequences; OM033041 to OM033132 for nrDNA ETS sequences; OM048521 to OM048611 for cpDNA matK sequences; and OM056558 to OM056660 for cpDNA rpl32-trnLUAG sequences.

All these accession numbers for newly generated sequences are available at Table 2 which also includes the voucher information of specimens.

Raw data are available in the Supplemental Files.

The following information was supplied regarding data availability:

Raw data are available in the Supplemental Files.

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
