# Peer review of "A phylogenetic analysis of Bromus (Poaceae: Pooideae: Bromeae) based on nuclear ribosomal and plastid data, with a focus on Bromus sect. Bromus"

_PeerJ, doi:10.7717/peerj.13884_

## Round 0.1 · original submission · Major Revisions

Both reviewers recommend Major revisions and I coincide with them. Results and Discussion need to be better organized according to the objectives of the paper to improve reading of the paper. Taxonomic and nomenclatural history might be synthesized better in a Table than in text, to facilitate reading. According to one of the reviewers objective 4 is not well justified, and additional analyses such as ABC or others might give more information on the hybrid origin of the species. For the extremely unresolved trees retrieved here more explanations and justifications and future research for resolving pylogenetic relationships in this group are needed. Previous research on the group should be added and explained better.

·

Basic reporting

The paper investigated the phylogenetic relationships among species of Bromus sect. Bromus and other sections of Bromus based on DNA sequences. I think that this study contributes to the understanding the phylogeographic history of the whole Poaceae, and it is adequately done technically. However, this paper still needs more revision. So, I would suggest partly restructuring and strengthening the text as major revisions.
Please refer the followings for detailed suggestions:
1. The authors discussed the evolutionary relationships of Bromus with nrDNA and cpDNA sequences, as the authors said, most of the relationships among the remaining species of Bromus sect. Bromus are unresolved due to low genetic differentiation. Thus, I want to know why the authors do not apply the morphological data and molecular data together. Especially, the authors can also try to select some markers with high evolutionary rates, such as single-copy nuclear markers.
2. The authors said that the phylogenetic tree within and among major lineages of Bromus was not the considerable incongruence from nuclear ribosomal and plastid data, respectively. Why the authors did not combine the nuclear ribosomal and plastid data to construct the phylogenetic tree?
3. Based on the current data in the present study, I think that they cannot well resolve the four objectives in this paper, especially the fourth objective. If you want to discuss the putative hybrid origin of species, I suggest you should try with model tests, such as ABC, IM, etc.
4. The authors should also carefully check the paper for the whole, especially spelling mistakes, reference format, and so on.

Experimental design

It is ok.

Validity of the findings

It is ok.

Additional comments

No.

·

Basic reporting

This is an interesting study about the phylogeny of Bromus. It includes several annual taxa that in some cases have not previously been included in molecular studies of the genus. The combined nrDNA (ETS, ITS) and the combined plastid regions (matK, trnL32-rpLUAG) show complex patterns, and only in some cases, the phylogenetic lineages involved are resolved.

The topic of the research is, in my opinion, quite suitable for PeerJ. Since I have worked on this exact topic, and know the species and the literature, can assess the importance of the results, which is interesting, and in some cases, confirm our previous unpublished findings.

Throughout the MS, clear and unambiguous, professional English is used.

The literature provided a sufficient background context. There are more than 160 items in the list of references. In many sentences, even 6-8 references are cited to support one idea or argument (e.g. Introduction). It is not necessary to cite all articles which have been published on the same or similar subject. I propose you decide which ones are really necessary and diminish the number of references.

Experimental design

Since Bromus are taxonomically critically, how did you make sure that specimens were correctly identified? Did you check all the voucher herbarium specimens?

I have no comments about the quality of the experimental design, but I suggest reducing the methods that are very detailed described. It could be shortened by only mentioning the literature that previously described the same methodology, and briefly explaining your possible modifications (if it is the case). The authors may benefit from that literature and approach.

The only knowledge gap identified is related to the missing sequences to confirm some of the findings, as the authors explain in several points.

Validity of the findings

To validate the findings presented in the MS, the data on which the conclusions are based, are provided as supplementary material (matrices), but original sequences are not yet (during my revision) available in GenBank. I thank you for providing the matrices, however, to be possible to check that MP, ML, and BI analyses report similar topologies, you must include the recovered trees as supplementary material to help potential readers understand your affirmations.

I suggest better organizing your results and discussion sections in four subsections following the four objectives of the paper differentiating your original findings and others you only corroborate.

The annual Bromus are taxonomically diverse. Although it is difficult to interpret in unresolved trees, checking only the resolved clades the monophyly of several species in your trees are not recovered (e.g. B.danthoniae, B.japonicus, B.racemosus, ). Assuming the problems arising in working with annuals that in several cases were originated by hybridization events or affected by introgressions, could you explain the situation of the conflictive taxa? It could be an interesting discussion point.

The accurate relationships were not supported in nuclear nor in plastid studies. These conflicts are potentially related to that phylogeny between diverged lineages is very difficult to infer if hybridization events exist, which in Bromus is widespread and deteriorates the underlying phylogenetic signal recovered.

A polytomy means that you do not have enough data to understand how the lineages are related, and although you produced a phylogeny you cannot draw any conclusions, only confirm rapid speciation. Sometimes means that multiple speciation events happened at the same time, and the daughter lineages are equally closely related to one another. Could you explain to the potential readers if there is evidence indicating what the case is?

About some of the “new insights into the evolutionary history of B.pumilio and B.gracillimus” the Alonso Ph.D. dissertation in 2015 showed, in the bases of the same nuclear molecular marker and the same sequences that the analysis you use. Please, could you explain it? In addition, it is not necessary to detail here all their taxonomic histories, because your objectives are different.

Many papers report the problems by using ITS in polyploids. As noted Bromus include many polyploids, originated by several hybridization events that could contain different ITS variants. Have your interpretations taken into account these important questions?

Additional comments

The manuscript is too long: e.g.. some 9000 words extend the discussion that is half of the main text (excluding references).

In my opinion, the discussion should be shortened, focused on the objectives of the study, and not extended unnecessarily by summarizing the results of previous related studies. The discussion could be more precisely structured and shortened. This preliminary version is difficult to follow.

In addition, I suggest that all the taxonomic/nomenclatural history, available in the literature, be summarized in a table, easier to follow, and eliminated from explanations in the discussion.

The several findings or explanations about results in the literature, related but out of the objectives of the study, must be also eliminated or substantially reduce to facility the potential readers to understand your own findings from the previous ones, published or unpublished.

---

## Round 0.2 · accepted · Accept

I appreciate the inclusion of Table 1 - it really helps to understand classifications of this section in Bromus. Probably, the production team of PeerJ will ask to increase the font size in some of the figures.

·

Basic reporting

It is ok.

Experimental design

The authors have revised it.

Validity of the findings

Right.

Additional comments

No.